# MGHF: Multi-Granular High-Frequency Perceptual Loss for Image Super-Resolution

## Abstract

An avalanche of innovations in perceptual loss has advanced the super-resolution (SR) literature, enabling the synthesis of realistic and detailed high-resolution images. However, most of these approaches rely on convolutional neural network (CNN)-based non-homeomorphic transforms, which result in information loss during guidance and often necessitate complex architectures and training procedures. To address these limitations—particularly the information loss and unwanted harmonics introduced by CNNs—we propose a diffeomorphic transform–based variant of a computationally efficient invertible neural network (INN) for a naive **M**ulti-**G**ranular **H**igh-**F**requency (MGHF-n) perceptual loss, trained on ImageNet. Building on this foundation, we extend the framework into a comprehensive variant (MGHF-c) that integrates multiple constraints to preserve, prioritize, and regularize information across several aspects: texture and style preservation, content fidelity, regional detail preservation, and joint content–style regularization. Information is prioritized through adaptive entropy-based pruning and reweighting of INN features, while a content–style consistency regularizer regulates excessive texture generation and ensures content fidelity. To capture intricate local details, we further introduce modulated PatchNCE on INN features as a local information preservation (LIP) objective. As another thread in the tapestry, we present the theoretical foundation, showing that (1) the LIP objective compels the SR network to maximize the mutual information between super-resolved and ground-truth modalities, and (2) a diffeomorphic transform–based perceptual loss enables more effective learning of the ground-truth distribution manifold compared to non-homeomorphic counterparts. Empirical results demonstrate that the proposed MGHF objective substantially improves both GAN- and diffusion-based SR algorithms across multiple evaluation metrics, and the code will be released publicly after the review process.

## 1 Introduction

Super-resolution (SR) aims to improve the detailed information in images degraded by down-sampling, blurring, noise, and various real-world distortions (Wang et al., 2020). Degraded images contain structural information but lack high-frequency information Zhang et al. (2024); Chen et al. (2022). Researchers employ various generative models (Wu et al., 2024a; Ledig et al., 2017; Lugmayr et al., 2020; Lu et al., 2022; Guo et al., 2022; Wei and Zhang, 2023) and objective functions (Johson et al., 2016; Zhang et al., 2018; Cheon et al., 2018; Kim et al., 2024a; Deng et al., 2019) to enhance high-frequency features in the SR problem Kim et al. (2016); Lugmayr et al. (2020); Wu et al. (2024a). The objective functions for SR can be categorized as perceptual (Zhang et al., 2018; Johson et al., 2016), content (Qin and Wang, 2024), style losses (Sajjadi et al., 2017), structural similarity measures (Wang et al., 2004; Singla et al., 2024), and frequency domain losses (Sims, 2020a; Cai et al., 2021). Among these categories, naive perceptual losses (Johson et al., 2016; Zhang et al., 2018) are widely used; however, while effective in capturing many characteristics of the source image, they fall short of preserving complete details due to the inherent information approximation (Yarotsky,

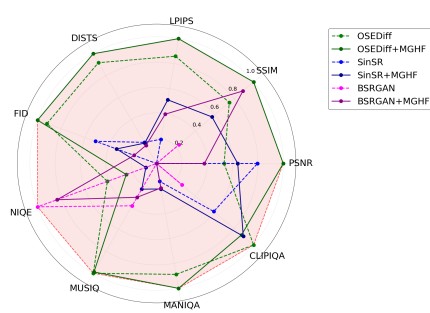 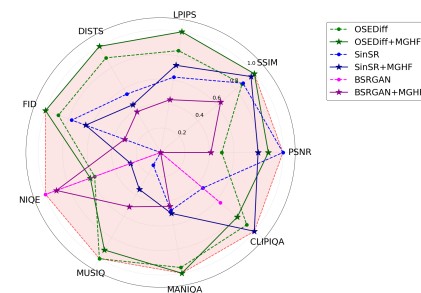

(a) Performance comparison on DrealSR       (b) Performance comparison on RealSR

Figure 1: Performance comparison of different super-resolution models with and without MGHF framework. (a) Results on the DrealSR (Wei et al., 2020) dataset showing the effectiveness of MGHF across different metrics. (b) Results on the RealSR (Cai et al., 2019) dataset demonstrate consistent improvements.The dotted line of each color represents the baseline model, and the solid line of the same color represents the baseline model with the MGHF framework.

2017; Achille and Soatto, 2018) and lossy nature of CNN operations (Jacobsen et al., 2018). In the SR literature (Deng et al., 2019; Fuoli et al., 2021; Zhang et al., 2018; Sims, 2020b), several variants of information approximation within the perceptual loss family have been implemented through diverse techniques such as quantization (Gray and Neuhoff, 1998), adversarial training (Liu et al., 2017), neural network feature extraction (Yarotsky, 2017; Lu et al., 2017; Tishby and Zaslavsky, 2015; Achille and Soatto, 2018), and feature enhancement (Dai et al., 2018). Some of these approximation approaches are: i) LPIPS (Zhang et al., 2018), which employs learned feature map weighting to align with human perception; ii) FDPL (Sims, 2020b), which applies quantization to discrete cosine transform (DCT) (Ahmed et al., 1974) coefficients, despite DCT's inherent lossless nature; iii) Fourier space loss (Fuoli et al., 2021), which shifts generation toward perceptually pleasing high-frequencies through adversarial training (Goodfellow et al., 2014); and iv) wavelet domain style transfer (Deng et al., 2019), which introduces feature enhancement through a selective wavelet filter. Moreover, since OSEDiff (Wu et al., 2024a) employs the LPIPS (Zhang et al., 2018) objective based on a non-homeomorphic CNN transform (Plastock, 1974) rather than a diffeomorphic (Earle and Eells, 1967) invertible neural network (INN) (Dinh et al., 2022), which introduces information loss and approximation errors in perceptual evaluation, our results in Fig. 1 show that the diffeomorphic transform-based multi-granular high-frequency (MGHF) framework effectively mitigates these issues and improves performance across several metrics.

Another inherent problem of several perceptual loss families during SR is the substantial complexity of the architecture design (Kim et al., 2024a; Rad et al., 2019) and training procedure (Ledig et al., 2017). For example, SRGAN (Ledig et al., 2017) employs a relatively straightforward perceptual loss (Johson et al., 2016) by using VGG (Simonyan and Zisserman, 2014) features, and requires unstable adversarial training (Goodfellow et al., 2014). SROBB (Rad et al., 2019) significantly increases complexity by introducing region-specific perceptual losses that process objects, backgrounds, and boundaries differently, requiring additional segmentation labels and specialized loss calculations for each semantic region. SR4IR (Kim et al., 2024a) presents the complex training methodology with its alternate training framework that switches between updating the SR network and the task network, combined with a specialized cross-quality patch mix data augmentation strategy. We propose a naive version of MGHF perceptual loss that maintains an efficient architecture while delivering effective results for the super-resolution task, addressing these complexity issues.

Perceptual losses (Johson et al., 2016; Zhang et al., 2018) trained on the VGG (Simonyan and Zisserman, 2014) or AlexNet (Krizhevsky et al., 2012) backbone in ImageNet (Deng et al., 2009) and stable diffusion (Rombach et al., 2022a) trained on billions of image-text pairs serve as important super-resolution priors (Wu et al., 2024a; Wang et al., 2024a). We introduce a novel high-frequency perceptual loss based on an invertible neural network (INN) trained on ImageNet as a new prior for SR. INNs have previously been utilized in image super-resolution and rescaling (Xiao et al., 2020) literature in ways distinct from our

approach. For instance, SRFlow (Lugmayr et al., 2020) employs INN-based normalizing flows (Rezende and Mohamed, 2015) to learn conditional distributions of high-resolution images given low-resolution inputs, while IRN (Xiao et al., 2020) explicitly models downscaling or upscaling as forward or inverse operations of an invertible network with Haar wavelet (Haar, 1910) transformation. HCFlow (Liang et al., 2021b) creates bijective mappings between HR-LR image pairs where high-frequency components are hierarchically conditional on low-frequency components through specially designed flow levels, and IARN (Pan et al., 2023) adapts the invertible framework by replacing Haar wavelet transforms with preemptive channel splitting and embedding position-aware scale encoding, enabling arbitrary rescaling factors within a single model while maintaining bidirectional invertibility. The authors (Wei et al., 2024) introduced invertible priors for image rescaling through Invertible Feature Recovery Modules (IFRM), which establish bijective transformations between quantized features obtained by VQGAN (Esser et al., 2021) and low-resolution images using coupling layers (Dinh et al., 2022). Extending this line of research, our work makes a distinct contribution by employing an INN trained on ImageNet as a super-resolution (SR) prior. Furthermore, we underscore a fundamental limitation of existing perceptual loss approaches (Johson et al., 2016; Zhang et al., 2018): the information loss and harmonic distortion introduced by non-homeomorphic transformations, such as MaxPooling and ReLU layers in AlexNet and VGG backbones, when computing widely adopted perceptual losses. This observation leads us to formulate the central research question: *Can a lossless, diffeomorphism-based super-resolution prior be established to facilitate more efficient and effective perceptual loss computation in comparison to conventional non-homeomorphic transforms?*

We propose a **m**ulti-**g**ranular **h**igh-**f**requency perceptual loss (MGHF) to overcome the aforementioned issues. The naive version, MGHF-n, serves as an effective invertible neural network (INN) prior trained on ImageNet to guide the super-resolution process. Building upon this foundation, our comprehensive version (MGHF-c) addresses the perception-distortion tradeoff (Blau and Michaeli, 2018) and improves the SR performance on several image quality metrics (Wang et al., 2023; Ke et al., 2021; Zhang et al., 2015) by both focusing and regularizing essential detail information alongside the INN prior. To achieve these goals, MGHF-c introduces an adaptive importance score based on normalized entropy to prioritize and select significant INN features, which are then processed through a multifaceted approach that incorporates a modulated PatchNCE (Zhan et al., 2022)-based local information preservation objective to maintain intricate details, while simultaneously preserving style and content information in the INN domain via Gram matrix and mean-squared loss, respectively. Additionally, to overcome unnecessary style transfer and preserve content information while guiding SR, we propose a correlation loss-based content-style consistency regularizer. Our experiments demonstrate that the proposed MGHF objective significantly improves the performance of three super-resolution algorithms: OSEDiff (Wu et al., 2024a), SinSR (Wang et al., 2024c), and BSR-GAN (Zhang et al., 2021), with the first two based on diffusion models, and the last on a GAN. Notably, in SinSR (Wang et al., 2024c), even our simpler variant, MGHF-n, outperforms both LPIPS (Zhang et al., 2018) and the naive perceptual loss (Johnson et al., 2016). Furthermore, our MGHF framework consistently outperforms several image enhancement (Zhu et al., 2024; Qin et al., 2024) approaches and remains robust across diverse degradation techniques (Wang et al., 2021c; Yue et al., 2022; Wang et al., 2021a; Yao et al., 2024) and scaling factors within OSEDiff. Also, our proposed INN feature extractor within the MGHF framework requires 41 times fewer parameters than the VGG (Simonyan and Zisserman, 2014)-based feature extractor typically used for calculating perceptual losses (Johson et al., 2016; Zhang et al., 2018). We term our approach the Multi-Granular High-Frequency (MGHF) perceptual loss, as it accounts for different levels of information—specifically style, content, and consistency—while the diffeomorphic transformation–based prior preserves high-frequency information during SR. The details of related works regarding SR methods and perceptual objectives are discussed in Appendix A.

## 2 METHODOLOGY

In this section, we introduce a diffeomorphic transform-based, multi-granular high-frequency perceptual objective for super-resolution and establish its theoretical advantages over non-homeomorphic alternatives.

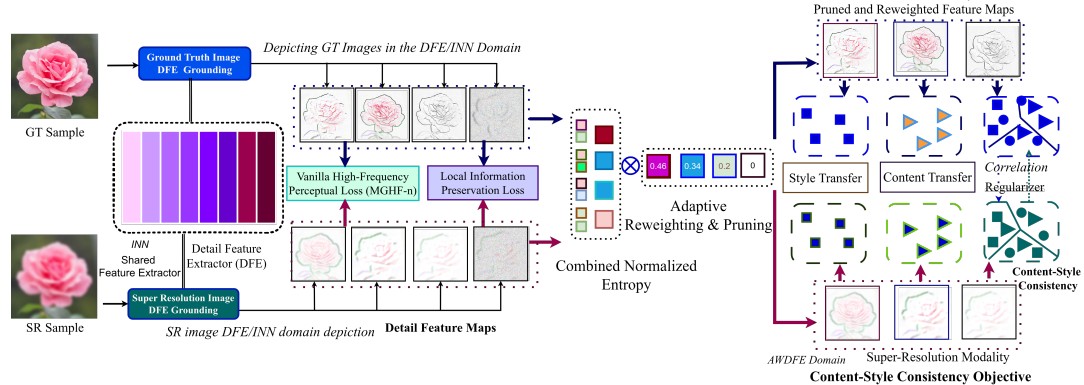

Figure 2: The depiction of proposed MGHF perceptual loss, where the detail feature extractor (DFE) is based on an invertible neural network. The vanilla high-frequency perceptual loss is calculated among feature maps of the DFE, while the content-style consistency loss is calculated from the most informative pruned and reweighted DFE feature maps.

We then present an invertible neural-network-based detail feature extractor (DFE) and its adaptive weighted variant (AWDFE), along with objectives for content–style consistency and local information preservation. We provide a concise overview here; full details appear in the Appendix.

## 2.1 DETAIL FEATURE EXTRACTOR

We propose a detail feature extractor (DFE), trained on ImageNet (Deng et al., 2009), to preserve texture, fine-grained detail, and content correspondence between super-resolution and ground-truth images. The DFE's backbone is an invertible neural network built from affine coupling layers (Dinh et al., 2022); a brief specification appears in Algorithm 1. The DFE adheres to the diffeomorphic principle, whereas conventional perceptual losses rely on CNN feature spaces that employ non-injective operations (MaxPooling, ReLU), which cause information loss and harmonic distortion (see Remark 1 and Corollary 1). The advantages of diffeomorphic over non-homeomorphic transforms are formalized in Proposition 1 and Theorem 1.

### 2.1.1 THEORY OF SUPERIORITY OF DIFFEOMORPHIC INN OVER CNN IN PERCEPTUAL LOSS CALCULATION

**Proposition 1.** [Information Preservation] *The use of non-homeomorphic transform-based perceptual loss results in information approximation, whereas a diffeomorphic transform-based perceptual loss preserves all frequency components during translation. Consequently, the latter facilitates superior performance in perceptual loss calculation.*

See proof in App. Sec. D.1. Using the first part of the proposition, the toy example in App. Sec. D.2 shows that a diffeomorphic transform preserves information, whereas a non-homomorphic transform does not.

**Remark 1.** [Information Loss in CNN] *The ReLU activation function and the MaxPooling operation are inherently non-injective mappings. As a consequence, they introduce irreversible information loss within perceptual loss frameworks that rely on feature representations extracted from AlexNet and VGG networks.*

These remarks can be explained by feature map visualization on Fig. 3, where deeper layers of VGG lost fine-grained details.

**Corollary 1.** [Frequency distortion by ReLU operation] *The output signal $y(t) = ReLU(\cos(\omega_0 t))$ contains frequency components at integer multiples of $\omega_0$ that were not present in the input signal $x(t) = \cos(\omega_0 t)$.*

See proof in App. Sec. D.5. This proof demonstrates that the ReLU operation introduces unwanted harmonics in a simple sinusoidal signal. While MaxPooling and ReLU cause the generalization capacity of CNNs (Brutzkus and Globerson, 2021; Banerjee et al., 2019), they also induce information loss and harmonic distortion—effects that can be detrimental in applications where strict information preservation is essential.

**Theorem 1.** *[Superiority of diffeomorphic INN over CNN in perceptual loss calculation]. Invertible Neural Networks (INNs) offer theoretical advantages over Convolutional Neural Networks (CNNs) when used as perceptual feature extractors. Formally, let $f : \mathbb{R}^n \to \mathbb{R}^n$ denote a diffeomorphic INN and $g : \mathbb{R}^n \to \mathbb{R}^m$ a standard CNN feature map with non-invertible operators (pooling, ReLU, strided convolutions). Then, the following contrasts hold:*

- ***Information conservation.** INN: $H(f(X)) = H(X)$ (entropy preserved due to bijectivity). CNN: $H(g(X)) < H(X)$ (irreversible compression due to non-invertibility).*

- ***Manifold preservation.** INN: diffeomorphic mappings preserve topology of the image manifold. CNN: distortion mappings collapse neighborhoods and destroy manifold structure.*

- ***Statistical equivalence.** INN: all statistical moments of $X$ are preserved in $f(X)$. CNN: higher-order moments are altered or lost.*

- ***Spectral completeness.** INN: full frequency spectrum preserved, including high-frequency details. CNN: effective low-pass filtering due to pooling and convolution kernels.*

- ***Gradient stability.** INN: Jacobians are well-conditioned ($\det J_f(x) \neq 0$). CNN: singular Jacobians induce unstable or vanishing gradients.*

- ***Distribution matching.** INNs theoretically achieve perfect distribution matching, whereas CNNs exhibit positive Wasserstein distance.*

We provide the proof in App. Sec. D.6. Our experimental results on App. Table 4 depict that the proposed diffeomorphic transform-based MGHF-n outperforms naive perceptual losses (Zhang et al., 2018; Johson et al., 2016).

Let $X_{GT}$ and $X_{LR}$ be the ground-truth and corresponding low-resolution image sample caused by down-sampling, blur, and real-world degradation. Any super-resolution method can transform $X_{LR}$ to $X_{SR}$. The DFE is used to extract detailed feature maps by:

$$\begin{aligned} \mathbf{G} &= \mathbf{DFE}(X_{GT}), & \mathbf{S} &= \mathbf{DFE}(X_{SR}), & where \\ \mathbf{G} &= \{G_1, G_2, \ldots, G_L\}, & \mathbf{S} &= \{S_1, S_2, \ldots, S_L\}, & L \text{ is the number of DFE feature maps.} \end{aligned} \tag{1}$$

The naive multi-granular high-frequency perceptual loss (MGHF-n) is calculated between DFE features of GT and SR images in the following way:

$$\mathcal{L}_{\text{MGHF-n}} = \mathcal{L}_{\text{MSE}}(\mathbf{G}, \mathbf{S}). \tag{2}$$

## 2.2 Adaptive and Weighted Detail Feature Extractor

The detail feature maps encompass various aspects of an image. However, some of the feature maps consist of less informative and redundant information. To overcome these issues and improve robustness (Correia et al., 2019; Niculae and Blondel, 2017) while calculating perceptual loss, we propose adaptive DFE filter weighting and pruning strategies that utilize the entropy calculation on DFE feature maps. The importance score ($I_{\text{combined}}(j)$) of all the extracted DFE feature maps is calculated using entropy, which enables us to select the most informative M feature maps from the L detailed feature maps by using Eq. 4. These M selected

---

**Algorithm 1** Pretraining of Detail Feature Extractor

---

**Require:** Invertible modules $\{\psi_k\}_{k=1}^K$, CNN modules $\{\mathcal{C}_l\}_{l=1}^L$, fully connected layers (FC), and convolution $\{\text{Conv}(3 \to N\ channel)\}$.
**Require:** ImageNet training set $\bar{\mathcal{Z}}$
1: **while** not converged **do**
2:     Sample $\bar{z} \sim \bar{\mathcal{Z}}$
3:     $z_0 \leftarrow \text{Conv}(\bar{z})$
4:     **for** $k \leftarrow 1$ **to** $K$ **do**
5:         $z_k \leftarrow \psi_k(z_{k-1})$
6:     **end for**
7:     $\hat{y}_1 \leftarrow z_K$
8:     **for** $l \leftarrow 1$ **to** $L$ **do**
9:         $\hat{y}_{l+1} \leftarrow \mathcal{C}_l(\hat{y}_l)$
10:     **end for**
11:     $y_{\text{score}} \leftarrow \text{Softmax}(\text{FC}(\hat{y}_{L+1}))$
12:     $\mathcal{L} \leftarrow \text{CrossEntropy}(y_{\text{score}}, y_{\text{class}})$
13:     Update Conv, $\{\psi_k\}$, $\{\mathcal{C}_l\}$, FC by descending $\nabla\mathcal{L}$
14: **end while**
15: **return** embedding $z_K$

---

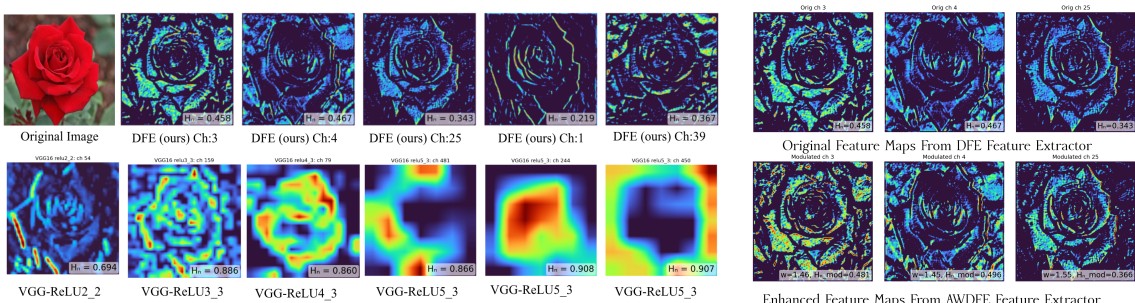

Figure 3: Visualization of feature maps of the detail feature extractor (DFE) and VGG. (Please zoom in on the figure for better perception.)

Figure 4: Visualization of original DFE and AWDFE feature maps.

feature maps are then weighted by introducing an adaptive weighting strategy in Eq. 5.

$$I_{\text{combined}}(j) = \frac{(1 - H_{\text{norm}}(G_j)) + (1 - H_{\text{norm}}(S_j))}{2}, \tag{3}$$

where $H_{norm}$ is the normalized entropy in the range $[0, 1]$; and $j = 1, 2, ..., L$ feature maps from DFE.

$$\begin{aligned} \mathcal{M} &= \{\text{indices of top } M \text{ feature maps}\}, \\ \hat{\mathbf{G}} &= \{G_i \mid i \in \mathcal{M}\} = \{G_{i_1}, \ldots, G_{i_M}\}, \text{ and similarly, } \hat{\mathbf{S}} \text{ is calculated.} \end{aligned} \tag{4}$$

$$\begin{aligned} w_i &= \left(1 + \alpha \cdot I_{\text{combined}}(i)\right)^\gamma, \quad i \in \mathcal{M}, \\ G_i^w &= w_i \cdot G_i, \quad S_i^w = w_i \cdot S_i, \quad i \in \mathcal{M}, \\ \mathbf{G}^w &= \{G_{i_1}^w, G_{i_2}^w, \ldots, G_{i_M}^w\}, \quad \mathbf{S}^w = \{S_{i_1}^w, S_{i_2}^w, \ldots, S_{i_M}^w\}, \end{aligned} \tag{5}$$

where $\hat{G}$ and $\hat{S}$ are the adaptive ground-truth and super-resolution pruned filters, respectively. $w_i$ is the importance score of $i$-th pruned feature map, and $\alpha$ and $\gamma$ are constant. By prioritizing and pruning the detail feature extractor's (DFE) outputs based on importance scores, we obtain the adaptive and weighted feature maps $\mathbf{G}^w$ and $\mathbf{S}^w$, constituting our AWDFE module.

## 2.3 CONTENT-STYLE CONSISTENCY

The content-style consistency objective preserves and regularizes the content and style features between ground-truth and super-resolution AWDFE features. While style and content information preservation is widely employed in super-resolution literature (Sajjadi et al., 2017; Cheon et al., 2018), we specifically utilize

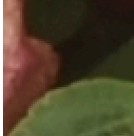 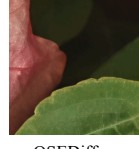 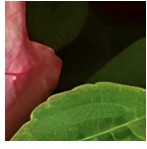 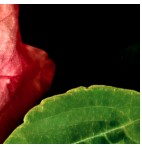 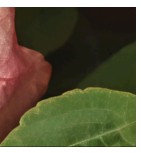 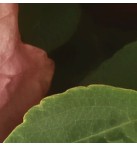

| Low Quality (LQ) | Zoomed LQ | OSEDiff (72.21; 0.4732) | OSEDiff + FlowIE (69.81; 0.4518) | OSEDiff + RAM (67.66; 0.4362) | OSEDiff + UnifyFormer (72.65; 0.4759) | OSEDiff + MGHF *(73.36; 0.4842)* |

Figure 5: Qualitative comparisons of different image enhancement methods in OSEDiff. Please zoom in for a better view. The values in the parenthesis are the quantitative result measured by (MUSIQ↑ (Ke et al., 2021); MANIQA↑ (Yang et al., 2022)). Our MGHF achieves better MUSIQ and MANIQA compared to others. However, the FlowIE (Zhu et al., 2024), RAM (Qin et al., 2024), and UnifyFormer (Yang et al., 2024) over-enhance the image.

style and content loss in the features of the AWDFE domain by applying the Gram matrix and mean squared error loss. We propose content-style consistency regularization by utilizing the correlation loss between SR and GT image pairs in the AWDFE domain. The total **c**ontent-**s**tyle **c**onsistency objective is denoted as ($\mathcal{L}_{\text{CSC}}$) in the following equation:

$$\mathcal{L}_{\text{CSC}} = \beta_1 \cdot \mathcal{L}_{\text{MSE}}(\mathbf{G}^w, \mathbf{S}^w) + \beta_2 \cdot \mathcal{L}_{\text{corr}}(\mathbf{G}^w, \mathbf{S}^w) + \beta_3 \cdot \mathcal{L}_{\text{Gram}}(\mathbf{G}^w, \mathbf{S}^w), \tag{6}$$

where $\mathcal{L}_{\text{corr}}$, $\mathcal{L}_{\text{Gram}}$, and $\mathcal{L}_{\text{MSE}}$ are content-style consistency regularizer, style, and content loss, respectively.

$$\mathcal{L}_{\text{corr}}(\mathbf{G}^w, \mathbf{S}^w) = 1 - \frac{1}{M} \sum_{i=1}^{M} \frac{\text{cov}(G_i^w, S_i^w)}{\sigma_{G_i^w} \cdot \sigma_{S_i^w}}, \quad \mathcal{L}_{\text{Gram}}(\mathbf{G}^w, \mathbf{S}^w) = \frac{1}{M} \sum_{i=1}^{M} \|\text{Gram}(G_i^w) - \text{Gram}(S_i^w)\|^2. \tag{7}$$

## 2.4 LOCAL INFORMATION PRESERVATION OBJECTIVE

Unpaired image-to-image domain translation (Zhu et al., 2017) is a well-known technique in the computer vision literature for transferring modalities. We assume super-resolution and ground truth modalities as two distinct modalities during the SR training procedure. To transfer GT to SR modality, we utilize the modulated patch-wise noise contrastive estimation (MoNCE) (Zhan et al., 2022) that effectively facilitates regional texture transfer. The proposed local information preservation (LIP) objective is calculated between DFE feature maps of SR and GT modalities, which can be depicted as:

$$
\begin{aligned}
\mathcal{L}_{LIP} &= \frac{1}{L} \sum_{k=1}^{L} \mathcal{L}_{MoNCE}(G_k, S_k), \\
&= \frac{1}{L} \sum_{k=1}^{L} \left\{ -\sum_{i=1}^{N_k} \log \left[ \frac{e^{(s_{ki} \cdot g_{ki}/\tau)}}{e^{(s_{ki} \cdot g_{ki}/\tau)} + Q(N_k - 1) \sum_{\substack{j=1 \\ j \neq i}}^{N_k} a_{ij}^k e^{(s_{ki} \cdot g_{kj}/\tau)}} \right] \right\},
\end{aligned}
\tag{8}
$$

where $L$ is the number of feature maps from DFE, each feature map is divided into $N_k$ patches, and each patch is projected into the embedding space. $a_{ij}$ is the weighting factor for a negative patch that is calculated through the Sinkhorn optimal transport plan (Cuturi, 2013). Thm. 2 demonstrates that our proposed LIP objective enhances information maximization between the GT and SR modalities. The further details of the mathematical formulations, and the parameter settings are provided in App. Sec. B.3, and App. Sec. C.3, respectively.

### 2.4.1 THEORY OF ENHANCEMENT OF INFORMATION MAXIMIZATION BY LIP OBJECTIVE

**Theorem 2.** [Information maximization between SR and GT modalities]

*The $\mathcal{L}_{LIP}$ objective provides a tighter lower bound on mutual information than standard InfoNCE.*

$$I(G; S) \geq \log N_k - \mathcal{L}_{LIP} \geq \log N_k - \mathcal{L}_{InfoNCE} \tag{9}$$

We present the detailed proof in App. Sec. D.7, while the experimental results in App. Table 6 highlight the significance of the LIP objective for super-resolution.

## 2.5 TOTAL OBJECTIVE

Our proposed MGHF-c framework optimizes the MGHF-n, content-style consistency, and local information preservation objectives. The overall objective can be defined as:

$$\mathcal{L}_{\text{MGHF-c}} = \Gamma_1 \cdot \mathcal{L}_{\text{MGHF-n}} + \Gamma_2 \cdot \mathcal{L}_{\text{CSC}} + \Gamma_3 \cdot \mathcal{L}_{\text{LIP}}, \tag{10}$$

where $\Gamma_1$, $\Gamma_2$ and $\Gamma_3$ are hyperparameters to balance the overall super-resolution process in multifarious granularity.

## 3 EXPERIMENT

### 3.1 EXPERIMENTAL SETUP

**Compared methods.** We analyze the performance of our proposed method with several super-resolution algorithms, e.g., StableSR-s200 (Wang et al., 2024b), RealSR-JPEG (Ji et al., 2020), DiffBIR-s50 (Lin et al., 2023), SeeSR-s50 (Wu et al., 2024b), OSEDiff (Wu et al., 2024a), PASD-s20 (Yang et al., 2023), ESRGAN (Wang et al., 2018), ResShift (Yue et al., 2023), SinSR (Wang et al., 2024c), BSRGAN (Zhang et al., 2021), SwinIR (Liang et al., 2021a), RealESRGAN (Wang et al., 2021c), DASR (Liang et al., 2022), and LDM (Rombach et al., 2022b). In addition, we evaluate MGHF against three contemporary image-enhancement approaches—RAM (Qin et al., 2024), FlowIE (Zhu et al., 2024), and UnifyFormer (Yang et al., 2024)—on the OSEDiff output.

**Metrics.** We employ PSNR, SSIM, DISTS (Ding et al., 2020), and LPIPS (Zhang et al., 2018) metrics for performance analysis on the testing dataset with reference images. Fréchet Inception Distance (FID) (Heusel et al., 2017) measures the distribution distance between ground-truth and generated images. Furthermore, we utilize five widely used non-reference image quality metrics to evaluate SR images' realism and semantic coherence: CLIPIQA (Wang et al., 2023), MUSIQ (Ke et al., 2021), MANIQA (Yang et al., 2022), QualiCLIP$^+$ (Agnolucci et al., 2024), and NIQE (Zhang et al., 2015).

### 3.2 EXPERIMENTAL RESULTS AND COMPARISON WITH STATE-OF-THE-ART

**Quantitative comparisons on real-world datasets.** We evaluate the performance of our proposed MGHF framework on three real-world datasets: RealSR (Cai et al., 2019), RealSet65 (Yue et al., 2023), and DrealSR (Wei et al., 2020). We investigate the image perceptual quality of MGHF compared with other state-of-the-art super-resolution algorithms in Table 1, and 2. As shown in Table 2, by applying our MGHF-n to SinSR, we achieve the best CLIPIQA (Wang et al., 2023) score among widely used GAN-, transformer-, and diffusion-based SR algorithms on the RealSR and RealSet65 datasets. We also analyze various reference and non-reference metrics of diffusion model-based approaches compared to ours on the DrealSRWei et al. (2020) and RealSRWu et al. (2024a) datasets in Table 1. In the RealSR and DrealSR datasets, OSEDiff+MGHF-c achieves the best LPIPS, DISTS, and FID scores. Furthermore, we visualize several samples with and without MGHF on the baseline methods in App. Fig. 8, and Fig. 9.

**Quantitative comparisons on synthetic datasets.** We investigate the reference-based fidelity metrics and non-reference-based image quality metrics in the ImageNet-Test (Deng et al., 2009) and DIV2K-Val (Agustsson and Timofte, 2017) datasets. From Table 3, the SinSR+MGHF-n method achieves the best MUSIQ and CLIPIQA scores and the second-best LPIPS score compared to the nine other SR approaches in the ImageNet-Test dataset, though SinSR+MGHF-n lags slightly in PSNR and SSIM metrics. We also find on the DIV2K-val dataset from Table 1 that MGHF-c significantly improves the performance on numerous metrics, e.g., SSIM, LPIPS, DISTS, FID, when applied to the OSEDiff, SinSR, and BSRGAN baseline models.

**Parameter and computational complexity of MGHF.** Our INN-based detailed feature extractor (DFE) provides substantial efficiency improvements compared to the conventional VGG16-based perceptual loss (Johson et al., 2016; Zhang et al., 2018) model. A comprehensive depiction of the time and space complexity of the

Table 1: Quantitative comparison with state-of-the-art SR methods ($4\times$ scaling) on both synthetic and real-world benchmarks. $s$ denotes the number of diffusion reverse steps. Highlighted skyblue, lightgreen, and orange rows are variants of the SR algorithm with our MGHF framework.

| Datasets | Methods | PSNR↑ | SSIM↑ | LPIPS↓ | DISTS↓ | FID↓ | NIQE↓ | MUSIQ↑ | MANIQA↑ | CLIPIQA↑ |
|---|---|---|---|---|---|---|---|---|---|---|
| DIV2K-Val | StableSR-s200 | 23.26 | 0.5726 | 0.3113 | 0.2048 | 24.44 | 4.7581 | 65.92 | 0.6192 | 0.6771 |
| | DiffBIR-s50 | 23.64 | 0.5647 | 0.3524 | 0.2128 | 30.72 | 4.7042 | 65.81 | 0.6210 | 0.6704 |
| | SeeSR-s50 | 23.68 | 0.6043 | 0.3194 | 0.1968 | 25.90 | 4.8102 | 68.67 | 0.6240 | 0.6936 |
| | PASD-s20 | 23.14 | 0.5505 | 0.3571 | 0.2207 | 29.20 | 4.3617 | 68.95 | 0.6483 | 0.6788 |
| | ResShift-s15 | 24.65 | 0.6181 | 0.3349 | 0.2213 | 36.11 | 6.8212 | 61.09 | 0.5454 | 0.6071 |
| | BSRGAN* | 22.67 | 0.5717 | 0.4428 | 0.2839 | 90.74 | 4.6398 | 58.92 | 0.4231 | 0.6268 |
| | BSRGAN*+MGHF-c | 23.27 | 0.5922 | 0.3910 | 0.2569 | 69.16 | 3.9963 | 62.54 | 0.4949 | 0.5875 |
| | SinSR-s1 | 24.41 | 0.6018 | 0.3240 | 0.2066 | 35.57 | 6.0159 | 62.82 | 0.5386 | 0.6471 |
| | SinSR +MGHF-c | 24.25 | 0.6100 | 0.3393 | 0.2202 | 50.78 | 5.6939 | 62.53 | 0.5208 | 0.6708 |
| | OSEDiff-s1 | 23.72 | 0.6108 | 0.2941 | 0.1976 | 26.32 | 4.7097 | 67.97 | 0.6148 | 0.6683 |
| | OSEDiff +MGHF-c | 24.27 | 0.6294 | 0.2824 | 0.1936 | 25.33 | 4.6985 | 68.5023 | 0.6200 | 0.6735 |
| | OSEDiff +RAM (Qin et al., 2024) | 17.61 | 0.5302 | 0.3655 | 0.2364 | 28.69 | 5.7877 | 65.47 | 0.5918 | 0.6410 |
| | OSEDiff +FlowIE (Zhu et al., 2024) | 22.20 | 0.6157 | 0.3692 | 0.2398 | 41.32 | 6.0190 | 64.18 | 0.5940 | 0.5711 |
| | OSEDiff +UnifyFormer (Yang et al., 2024) | 23.76 | 0.6154 | 0.2982 | 0.2010 | 26.61 | 4..8104 | 68.54 | 0.6071 | 0.5798 |
| DrealSR | StableSR-s200 | 28.03 | 0.7536 | 0.3284 | 0.2269 | 148.98 | 6.5239 | 58.51 | 0.5601 | 0.6356 |
| | DiffBIR-s50 | 26.71 | 0.6571 | 0.4557 | 0.2748 | 166.79 | 6.3124 | 61.07 | 0.5930 | 0.6395 |
| | SeeSR-s50 | 28.17 | 0.7691 | 0.3189 | 0.2315 | 147.39 | 6.3967 | 64.93 | 0.6042 | 0.6804 |
| | PASD-s20 | 27.36 | 0.7073 | 0.3760 | 0.2531 | 156.13 | 5.5474 | 64.87 | 0.6169 | 0.6808 |
| | ResShift-s15 | 28.46 | 0.7673 | 0.4006 | 0.2656 | 172.26 | 8.1249 | 50.60 | 0.4586 | 0.5342 |
| | BSRGAN* | 26.79 | 0.7580 | 0.4027 | 0.2839 | 224.89 | 5.9202 | 53.18 | 0.4334 | 0.6067 |
| | BSRGAN*+MGHF-c | 27.66 | 0.7895 | 0.3454 | 0.2497 | 198.54 | 5.9792 | 58.20 | 0.4956 | 0.5552 |
| | SinSR-s1 | 28.36 | 0.7515 | 0.3665 | 0.2485 | 170.57 | 6.9907 | 55.33 | 0.4884 | 0.6383 |
| | SinSR +MGHF-c | 28.10 | 0.7759 | 0.3334 | 0.2488 | 185.78 | 6.8817 | 57.51 | 0.4967 | 0.6813 |
| | OSEDiff-s1 | 27.92 | 0.7835 | 0.2968 | 0.2165 | 135.30 | 6.4902 | 64.65 | 0.5899 | 0.6963 |
| | OSEDiff +MGHF-c | 28.87 | 0.8057 | 0.2713 | 0.2088 | 132.52 | 6.8203 | 64.27 | 0.6012 | 0.6995 |
| | OSEDiff +RAM (Qin et al., 2024) | 18.31 | 0.6502 | 0.3928 | 0.2717 | 140.96 | 7.1188 | 63.01 | 0.5734 | 0.6957 |
| | OSEDiff +FlowIE (Zhu et al., 2024) | 24.57 | 0.7805 | 0.2882 | 0.2347 | 161.88 | 8.0641 | 61.39 | 0.5714 | 0.5806 |
| | OSEDiff +UnifyFormer (Yang et al., 2024) | 27.97 | 0.7889 | 0.2928 | 0.2190 | 137.32 | 6.5703 | 65.33 | 0.5823 | 0.6180 |
| RealSR | StableSR-s200 | 24.70 | 0.7085 | 0.3018 | 0.2288 | 128.51 | 5.9122 | 65.78 | 0.6221 | 0.6178 |
| | DiffBIR-s50 | 24.75 | 0.6567 | 0.3636 | 0.2312 | 128.99 | 5.5346 | 64.98 | 0.6246 | 0.6463 |
| | SeeSR-s50 | 25.18 | 0.7216 | 0.3009 | 0.2223 | 125.55 | 5.4081 | 69.77 | 0.6442 | 0.6612 |
| | PASD-s20 | 25.21 | 0.6798 | 0.3380 | 0.2260 | 124.29 | 5.4137 | 68.75 | 0.6487 | 0.6620 |
| | ResShift-s15 | 26.31 | 0.7421 | 0.3460 | 0.2498 | 141.71 | 7.2635 | 58.43 | 0.5285 | 0.5444 |
| | BSRGAN* | 24.02 | 0.6830 | 0.3949 | 0.2716 | 218.79 | 5.1710 | 59.67 | 0.4424 | 0.6350 |
| | BSRGAN*+MGHF-c | 24.95 | 0.7207 | 0.3416 | 0.2463 | 185.26 | 5.2761 | 64.49 | 0.5314 | 0.5572 |
| | SinSR-s1 | 26.28 | 0.7347 | 0.3188 | 0.2353 | 135.93 | 6.2872 | 60.80 | 0.5385 | 0.6122 |
| | SinSR +MGHF-c | 25.82 | 0.7397 | 0.3069 | 0.2419 | 148.88 | 5.9970 | 62.94 | 0.5430 | 0.6792 |
| | OSEDiff-s1 | 25.15 | 0.7341 | 0.2921 | 0.2128 | 123.49 | 5.6476 | 69.09 | 0.6326 | 0.6693 |
| | OSEDiff +MGHF-c | 26.01 | 0.7418 | 0.2731 | 0.2057 | 111.54 | 5.6058 | 68.32 | 0.6419 | 0.6673 |
| | OSEDiff +RAM (Qin et al., 2024) | 16.84 | 0.6025 | 0.3601 | 0.2666 | 134.34 | 6.0761 | 68.34 | 0.6200 | 0.6761 |
| | OSEDiff +FlowIE (Zhu et al., 2024) | 23.19 | 0.7310 | 0.2784 | 0.2219 | 148.45 | 7.2668 | 64.33 | 0.5923 | 0.5279 |
| | OSEDiff +UnifyFormer (Yang et al., 2024) | 25.18 | 0.7387 | 0.2862 | 0.2157 | 124.47 | 5.5983 | 69.56 | 0.6297 | 0.5666 |

components in the MGHF-c objective and the VGG-16 feature extractor is provided in App. Table 5.

**Comparisons with image enhancement methods.** We qualitatively and quantitatively compare our MGHF objective with different image enhancement methods in Table 1 and Fig. 5. Experimental results show the performance gain of MGHF over some image enhancement approaches on OSEDiff.

**Robustness under real-world degradations and blind SR.** We conduct a quantitative assessment of the MGHF objective under different real-world degradation methods, e.g., haze, rain, ISP signal, noise, JPEG, etc., in App. Table 8, evidencing consistent robustness.

**Ablation study of different objective functions in MGHF.** We thoroughly investigate the importance of each loss function in the MGHF-c objective. A comprehensive experiment of each objective and its optimal hyperparameter choice is discussed in App. Table 6, and Sec. C.

## 4 CONCLUSION AND LIMITATION

Despite the Cambrian explosion of perceptual objectives in the super-resolution (SR) literature, diffeomorphism-based approaches to preserving high-frequency content remain largely unexplored. This manuscript identifies limitations of existing non-homeomorphic transform-based perceptual losses and demonstrates the theoretical and empirical advantages of diffeomorphic transforms. We also affirm that tighter lower bounds on mutual information between ground truth and SR modalities enhance SR.

The MGHF framework demonstrates consistent improvements across OSEDiff, SinSR, and BSRGAN. In the next stage, we aim to generalize MGHF to flow-, autoregressive-, transformer-, and neural-operator-based super-resolution architectures. In CLIPIQA evaluations, our approach—along with several enhancement baselines—shows a marginal degradation on OSEDiff. This observation motivates an analysis of biases in CLIPIQA (Agnolucci et al., 2024; Miyata, 2023), which we address using the QualiCLIP[+] metric on test sets (Wei et al., 2020; Agustsson and Timofte, 2017; Cai et al., 2019) (see App. Sec. C.2 and Table 9).

## REPRODUCIBILITY STATEMENT

To facilitate reproducibility of our empirical results and findings, we intend to make our code publicly available in the final version. We describe all mathematical and algorithmic details necessary to reproduce our results throughout this paper. In Sec. 2, Sec. D, we outline the theoretical basis and mathematical framework for our method. Furthermore, we provide pseudocode for our method in Algorithm 1. For our theoretical contributions, we offer detailed proofs of theorems and propositions in Sec. D, Sec. D.1, Sec. D.5, Sec. D.6, and Sec. D.7. We provide experimental details in Sec. 3,and Sec. C. We have utilized the large language model (LLM) for grammatical correction of the manuscript and information collection from online sources.

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

# MGHF: Multi-Granular High-Frequency Perceptual Loss for Image Super-Resolution

In the appendix, we provide the following materials:

- Related works regarding different image super-resolution and perceptual objectives on super-resolution.
- Elaboration of the invertible neural network-based detail feature extractor.
- Preliminary discussion of local information preservation objective, therefore, we discussed PatchNCE and Modulated PatchNCE.
- Visual comparisons of real-world and synthetic samples are shown under a $4\times$ scaling factor.
- Ablation study in multifarious perspectives.
- Mathematical foundation and proofs of our proposed approach.

## A APPENDIX: RELATED WORKS

### A.1 IMAGE SUPER-RESOLUTION

Super-resolution is a well-known low-level computer vision problem widely used in many applications (Wang et al., 2020; Dong et al., 2015), such as surveillance (Aakerberg et al., 2022), medical imaging (Qiu et al., 2024), gaming (Dong et al., 2022), virtual reality (Spagnolo et al., 2023), photography (Park et al., 2023), face recognition (Chen et al., 2020), etc. After the evolution of AlexNet (Krizhevsky et al., 2012), researchers implemented deep learning-based super-resolution approaches (Dong et al., 2015; Johnson et al., 2016). Following that, the generative adversarial network (GAN) evolved, and the GAN-based SR algorithms (Ledig et al., 2017; Wang et al., 2018; Zhang et al., 2021) were mainstream in the computer vision community (Dong et al., 2015). The SR-GAN (Ledig et al., 2017), ESRGAN (Wang et al., 2018), and RankSRGAN (Zhang et al., 2019) are some well-known GAN-based super-resolution algorithms. The invertible neural network-based SRFlow (Lugmayr et al., 2020) outperformed the GAN-based SR algorithms in 2020. Furthermore, the transformer (Vaswani et al., 2017) is the dominant network for natural language processing, image classification, and detection, which facilitates researchers to implement the transformer in super-resolution (Lu et al., 2022). Additionally, the denoising diffusion model outperforms the GAN in various perceptual metrics within the generative computer vision field (Dhariwal and Nichol, 2021). The first denoising diffusion model-based SR algorithm was introduced in 2021 (Saharia et al., 2022). However, these early diffusion-based SR algorithms (Saharia et al., 2022; Yue et al., 2023) initially faced challenges with slow sampling speeds and required many inference steps. Recently, researchers (Wang et al., 2024c; Wu et al., 2024a; Zhang et al., 2024) have successfully developed diffusion-based super-resolution methods that can operate in a single step. Autoregressive models and neural operator-based SR algorithms (Guo et al., 2022; Wei and Zhang, 2023; Liu and Tang, 2024) have also been successfully employed in the SR domain. Our paper introduces a novel family of perceptual loss objectives that improve several state-of-the-art SR algorithms (Wu et al., 2024a; Wang et al., 2024c; Zhang et al., 2021) across different metrics.

### A.2 PERCEPTUAL OBJECTIVES IN SUPER-RESOLUTION

In the super-resolution literature, various perceptual losses have been proposed to improve realistic texture and edge generation. Initial works utilized a pretrained VGG network (Simonyan and Zisserman, 2014), alongside multiple training strategies (Zhang et al., 2018) and the inclusion of adversarial loss (Ledig et al., 2017). Wavelet domain style transfer (Deng et al., 2019) has improved the perception-distortion trade-off in SR by enhancing low-frequency features and transferring style into the wavelet domain. Frequency domain perceptual loss emphasizes several frequency bands of an image to depict its perceptual quality better (Sims, 2020a). Targeted perceptual loss has been applied in SR, utilizing semantic information (object, background, boundary labels) across different image regions to compute perceptual loss and enhance texture and edge quality (Rad et al., 2019). Furthermore, Fourier loss introduces adversarial losses directly in Fourier space to enable perception-oriented SR, allowing a smaller network to achieve comparable perceptual

quality (Fuoli et al., 2021). Task-driven perceptual (TDP) loss guides SR networks in restoring high-frequency details relevant to specific recognition tasks (Kim et al., 2024b). The authors (Mechrez et al., 2019) demonstrate that contextual loss approximates KL divergence as a statistical comparison tool for a more effective super-resolution strategy. The authors of EnhanceNet (Sajjadi et al., 2017) argue that traditional SR methods optimize for pixel-wise accuracy (PSNR) but tend to produce blurry images during SR. Consequently, the authors propose combining adversarial training with perceptual loss and a novel texture-matching loss to facilitate the generation of more realistic textures. Perceptual content losses (Cheon et al., 2018) utilize various perceptual loss functions, including discrete cosine transform coefficient loss and differential content loss, in conjunction with adversarial networks for super-resolution. The SSDNet (Zhao et al., 2023b) maps RGB and depth features to spherical space for improved feature decomposition, then fuses and refines the information to achieve depth map super-resolution. The Discrete Cosine Transform (DCT)-based perceptual loss emphasizes structural information that is sensitive to the human visual system (Sekhavaty-Moghadam et al., 2024). FreqNet (Cai et al., 2021) uses the DCT to learn and reconstruct high-frequency details, the spatial extraction network (SEN), which extracts and transforms spatial features from the low-resolution input image into frequency-domain features, and a frequency reconstruction network (FRN), which reconstructs the high-frequency details. Our MGHF framework prioritizes, preserves, and regularizes multi-granular information, including details, style, content, and regional characteristics, during super-resolution.

In the subsequent section, we will discuss the different components of the MGHF framework: the invertible neural network-based detailed feature extractor, adaptive filter pruning, and reweighting of the detailed features. We will also address our content-style consistency approach that preserves and regularizes content and style information in the INN domain.

## B  Appendix: Elaboration of Different Components in MGHF

### B.1  Detail Feature Extractor

We utilize an invertible neural network (INN) to capture high-frequency detail features in our proposed MGHF framework. In the NICE paper (Dinh et al., 2015), researchers first proposed the INN concept. The authors of RealNVP (Dinh et al., 2022) subsequently developed the *affine coupling layer*, which enabled more efficient and straightforward data inversion. Utilizing 1×1 invertible convolution, the Glow paper (Kingma and Dhariwal, 2018) demonstrated generation of realistic high-resolution images. INNs have been applied beyond generation—they've improved classification tasks through superior feature extraction capabilities and information-preserving properties (Finzi et al., 2019). Moreover, the INN-based detail feature extractor is also used in visible-infrared image fusion (Zhao et al., 2023a) and sensor fusion (Sami et al., 2025) literature. Let $X_{GT}$ and $X_{LR}$ be the ground-truth and corresponding low-resolution image sample caused by down-sampling, blur, and real-world degradation. Any super-resolution method transforms $X_{LR}$ to $X_{SR}$. The DFE is used to extract detailed feature maps by:

$$\mathbf{G} = \mathbf{DFE}(X), \qquad \mathbf{S} = \mathbf{DFE}(X), \qquad where$$
$$\mathbf{G} = \{G, G, \ldots, G\}, \quad \mathbf{S} = \{S, S, \ldots, S\}, \qquad L \text{ is the number of DFE feature maps.} \tag{11}$$

where $G$ and $S$ represent detail features extracted from the ground-truth and super-resolution images, respectively. The invertible module in the DFE consists of affine coupling layers (Dinh et al., 2022). The illustration of the invertible module is in Figure 6. In this figure, $\psi_{I,l}^S[1:c]$ is the first $c$ channels of the input feature at the $l$-th invertible layer, where $l = 1, \cdots, L$. The arbitrary mapping functions in each invertible layer are: $\mathcal{I}_1$, $\mathcal{I}_2$, and $\mathcal{I}_3$. We utilize the shallow diffeomorphic module (Earle and Eells, 1967) as an arbitrary mapping function in the invertible module. Moreover, $G = \psi_{I,L}(X_{GT})$. Finally, the extraction of $S = \mathbf{DFE}(X_{SR}) = \psi_{I,L}(X_{SR})$ can be calculated in the same way as $G$.

### B.2  PatchNCE Objective

We introduce a local information preserving (LIP) objective in our MGHF framework. The building block of MGHF is the modulated PatchNCE objective. To understand this, we will first discuss the naive PatchNCE objective. The CUT (Park et al., 2020) was one of the pioneering works that introduced a method to maximize the mutual information between the input patch and the corresponding output patch to preserve the semantic content in an unpaired I2I translation (Zhu et al., 2017) scheme by utilizing a contrastive learning framework. After that, several research studies (Zhan et al., 2022; Sami et al., 2023; Wang et al., 2021b) have improved the CUT architecture. The PatchNCE objective maximizes the mutual

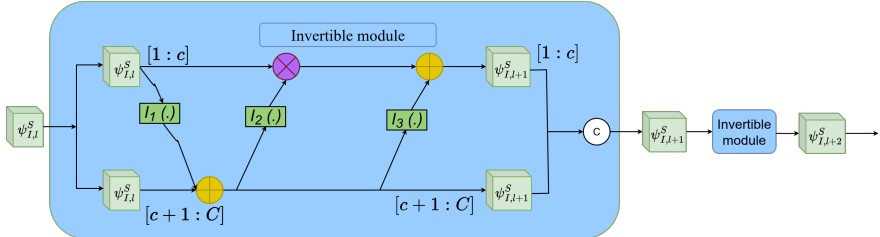

Figure 6: The architecture of the invertible module in the detail feature extractor (DFE) when calculating the multi-granular high-frequency perceptual framework. The DFE consists of $L$ cascaded invertible modules (Dinh et al., 2022). Each invertible module has an affine coupling layer consisting of scaling and translation functions and a $\odot$ Hadamard product. We use shallow diffeomorphic modules to conduct the scaling and translation operations. Each invertible module contains three shallow diffeomorphic modules.

information, $I(X,Y) = H(X) - H(X|Y)$, which is equivalent to minimizing the conditional entropy $H(X|Y)$. The PachNCE objective can be denoted as:

$$\mathcal{L}_{Patch-NCE}(X,\bar{Y}) = -\sum_{i=1}^{N} log[\frac{e^{(\bar{y}.x/\tau)}}{e^{(\bar{y}.x/\tau)} + \sum_{j=1}^{N} e^{(\bar{y}.x/\tau)}}], \tag{12}$$

where $\tau$ is a temperature parameter, and $\bar{Y}$ and $X$ are the generated target domain and ground truth images, respectively. $X = [x_1, x_2, \ldots, x_N]$ and $\bar{Y} = [\bar{y}_1, \bar{y}_2, \ldots, \bar{y}_N]$ represent encoded feature vectors from the 1st, 4th, 8th, 12th, and 16th layers of the encoder. Afterward, these features are passed through a two-layer MLP network (Rosenblatt, 1957; Park et al., 2020; Zhan et al., 2022). Unlike PatchNCE, we introduce feature maps from every layer of the detail feature extractor while calculating our proposed LIP objective B.4.

In the standard PatchNCE objective, N-class classification is performed where the anchor applies the same contrastive force on all $N-1$ negative patches, which is often too stringent and detrimental for convergence (Zhan et al., 2022). To address this issue, we utilize the modulated contrast NCE loss (Zhan et al., 2022) when calculating our local information preservation loss.

## B.3 Modulated Patch-wise Noise Contrastive Estimation Objective

In the contrastive learning literature, the hardness of negative samples has been addressed adequately in (Robinson et al., 2020; Wang et al., 2021b; Kalantidis et al., 2020). In contrastive learning literature, hard negatives have facilitated the learning of data representations (Robinson et al., 2020). The hardness of negative patches in unpaired image translation is defined by their similarity to the query (Zhan et al., 2022). As shown in Eq. 13, hard negative weighting defines the similarity between a negative sample $x_j$ and an anchor $\bar{y}_i$:

$$a_{ij} = \frac{e^{(\bar{y}.x/\beta)}}{\sum_{j=1}^{N} e^{(\bar{y}.x/\beta)}}, \tag{13}$$

where $\beta$ is the weighting temperature parameter. The modulated NCE objective employs reweighing procedures by implementing the constraint represented by the following equation:

$$\sum_{i=1}^{N} a_{ij} = 1, \sum_{j=1}^{N} a_{ij} = 1; i, j \in [1, N]. \tag{14}$$

Considering the optimal transport (Peyré et al., 2019), Eq. 15 provides the primary framework, subject to the constraints of Eq. 14.

$$\min_{a, i, j \in [1,N]} [\sum_{i=1}^{N} \sum_{\substack{j=1 \\ j \neq i}}^{N} a_{ij}.e^{\bar{y}.x/\tau}], \tag{15}$$

$$\min_T \langle C, T \rangle \ \ s.t \ \ \langle T \overrightarrow{1} \rangle = 1, \ \langle T^T \overrightarrow{1} \rangle = 1, \tag{16}$$

where $\langle C, T \rangle$ is the inner product of the cost matrix ($C$) and transport plan ($T$). In the unpaired I2I network and local information preservation objective, the cost matrix is $e^{\bar{y}.x/\beta}$ where $i \neq j$; if $i = j$ then $C_{ij} = \infty$. The Sinkhorn (Cuturi, 2013) algorithm is applied to Eq. 16 for calculating the optimal transport plan. Furthermore, while calculating the modulated contrastive objective in our LIP loss, we use every layer of feature maps of the detail feature extractor. The examples of vanilla and modulated contrast are depicted in Figure 7(a). and Figure 7(b). The MoNCE objective($\mathcal{L}_{MoNCE}$) can be expressed as:

$$\mathcal{L}_{MoNCE} = -\sum_{i=1}^{N} log[\frac{e^{(\bar{y}.x/\tau)}}{e^{(\bar{y}.x/\tau)} + Q(N-1)\sum_{j=1}^{N} a_{ij} e^{\bar{y}.x/\tau}}], \tag{17}$$

where $Q$ denotes the weight of negative terms in the denominator and typically $Q = 1$.

### B.4 LOCAL INFORMATION PRESERVATION OBJECTIVE

We assume super-resolution and ground truth modalities are two distinct modalities during the training. To transfer GT to SR modality, we utilize the modulated patch-wise noise contrastive estimation (MoNCE) (Zhan et al., 2022) that effectively facilitates regional texture transfer. The proposed local information preservation objective is calculated between the detail feature extractor (DFE) feature maps of SR and GT modalities, which can be depicted as:

$$
\begin{aligned}
\mathcal{L}_{LIP} &= \frac{1}{L} \sum_{k=1}^{L} \mathcal{L}_{MoNCE}(G_k, S_k), \\
&= \frac{1}{L} \sum_{k=1}^{L} \left\{ -\sum_{i=1}^{N} \log \left[ \frac{e^{(s \cdot g/\tau)}}{e^{(s \cdot g/\tau)} + Q(N_k - 1) \sum_{\substack{j=1 \\ j \neq i}}^{N} a_{ij}^k e^{(s \cdot g/\tau)}} \right] \right\},
\end{aligned} \tag{18}
$$

where $L$ is the number of feature maps from DFE, each feature map is divided into $N_k$ patches, and each patch is projected into the embedding space. $a_{ij}$ is the weighting factor for a negative patch that is calculated through the Sinkhorn optimal transport plan (Cuturi, 2013). The mathematical framework of MoNCE (Zhan et al., 2022) is elaborately described in **??**.

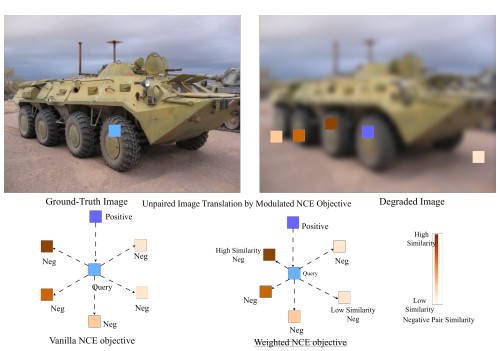

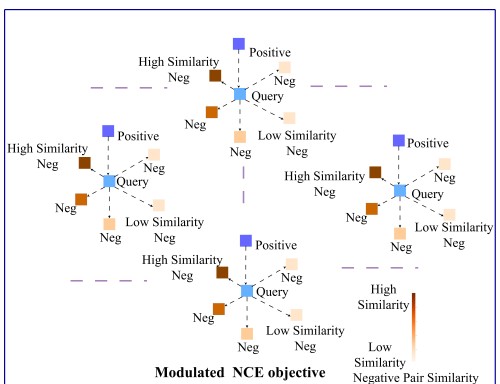

(a) Vanilla and weighted contrastive objective (Zhan et al., 2022).

(b) Modulated contrastive objective (Zhan et al., 2022).

Figure 7: The depiction of modulated contrastive objective (Zhan et al., 2022), which is utilized in our proposed local information preservation objective for image super-resolution.

## C APPENDIX: EXPERIMENT AND RESULTS

**Training Details.** While training different models with the MGHF objective, we adopt the same model architecture and parameter setup as their corresponding original baselines (Wang et al., 2024c; Wu et al., 2024a; Zhang et al., 2021). For all models, we follow the real-world degradation pipeline (Wang et al., 2021c; Zhang et al., 2021). We maintain the original training protocols and datasets for each model: SinSR is trained on ImageNet (Deng et al., 2009), while OSEDiff uses the LSDIR (Li et al., 2023) dataset combined with the first 10K face images from FFHQ (Karras et al., 2019). We train OSEDiff+MGHF and SinSR+MGHF following the same procedure as in the original OSEDiff and SinSR papers, respectively. For training both BSRGAN and BSRGAN+MGHF from scratch, we use the LSDIR (Li et al., 2023) dataset and the first 10K face images from FFHQ (Karras et al., 2019) for five epochs.

In the total objective equation 10, we determine the optimal values of $\Gamma_1$, $\Gamma_2$, and $\Gamma_3$ to be 2, 2, and $8 \times 10^{-2}$, respectively. In Eq. (3), we determine the optimal values of $\beta_1$, $\beta_2$, and $\beta_3$ to be $6 \times 10^{-3}$, $10^{-1}$, and $5 \times 10^{-4}$, respectively. Based on empirical observations, we set $\alpha = 1.1$ and $\gamma = \sqrt{2}$ in Eq. (5). A more detailed analysis of this objective is provided in Sec. C.1 and illustrated in Sec. C.1. Our experiments are conducted on two workstations, each equipped with two NVIDIA RTX A6000 GPUs.

**Training Detail Feature Extractor.** We train our detail feature extractor (based on an invertible neural network (Zhao et al., 2023a)) alongside convolutional and fully-connected layers to calculate MGHF perceptual loss. Initially, we use a convolutional block (He et al., 2016) to expand the image feature map from 3 to $N (= 128)$. The N channel of an image then passes through an invertible neural network. We take the output from the detail feature extractor to calculate MGHF-n perceptual loss. This network is trained on the ImageNet (Deng et al., 2009) dataset. We train this model for 20 epochs with a learning rate of 5e-4 with a batch size of 32 and an exponential scheduler with a factor of 0.95 every 5000 iterations. This model is optimized by Adam (Kingma, 2014) optimizer.

**Qualitative comparisons.** We visually compare four samples with and without the use of MGHF on OSEDiff (Wu et al., 2024a), SinSR (Wang et al., 2024c), and BSRGAN (Zhang et al., 2021) in Fig. 8 and Fig. 9. From these comparisons, we deduce that MGHF captures more details than the corresponding baseline approaches.

| Methods | Datasets | | | |
| --- | --- | --- | --- | --- |
| | *RealSR*[†] | | *RealSet65* | |
| | CLIPIQA↑ | MUSIQ↑ | CLIPIQA↑ | MUSIQ↑ |
| ESRGAN | 0.2362 | 29.048 | 0.3739 | 42.369 |
| RealSR-JPEG | 0.3615 | 36.076 | 0.5282 | 50.539 |
| BSRGAN | 0.5439 | **63.586** | 0.6163 | **65.582** |
| SwinIR | 0.4654 | 59.636 | 0.5782 | 63.822 |
| RealESRGAN | 0.4898 | 59.678 | 0.5995 | 63.220 |
| DASR | 0.3629 | 45.825 | 0.4965 | 55.708 |
| LDM-15 | 0.3836 | 49.317 | 0.4274 | 47.488 |
| ResShift-15 | 0.5958 | 59.873 | 0.6537 | 61.330 |
| *SinSR-1* | **0.6887** | 61.582 | **0.7150** | 62.169 |
| *SinSR-1 + MGHF-n* | **0.7240** | **61.897** | **0.7405** | **63.966** |

[†] RealSR is preprocessed with similar procedure as SinSR.

Table 2: Quantitative comparison among different super-resolution models on two real-world datasets. The best and the second best results among the SR methods are highlighted in **red** and **blue** colors, respectively.

### C.1 ABLATION STUDY

**Effectiveness of naive multi-granular high-frequency (MGHF-n) perceptual loss.** The effectiveness of the proposed MGHF-n perceptual loss can be deduced from the quantitative comparison in Tables 2, 3, and 4. All these results depict the efficacy of MGHF-n in the SinSR algorithm. Specifically, CLIPIQA (Wang et al., 2023) is significantly improved by the naive MGHF objective. Also, from Table 6, we observe that MGHF-n improves PSNR, SSIM, and LPIPS when applied to OSEDiff.

**Effectiveness of content-style consistency (CSC) and local information preservation (LIP) objective in MGHF.**

| Methods | Metrics | | | | |
|---|---|---|---|---|---|
| | PSNR↑ | SSIM↑ | LPIPS↓ | CLIPIQA↑ | MUSIQ↑ |
| ESRGAN | 20.67 | 0.448 | 0.485 | 0.451 | 43.615 |
| RealSR-JPEG | 23.11 | 0.591 | 0.326 | 0.537 | 46.981 |
| BSRGAN | 24.42 | 0.659 | 0.259 | 0.581 | **54.697** |
| SwinIR | 23.99 | 0.667 | 0.238 | 0.564 | 53.790 |
| RealESRGAN | 24.04 | 0.665 | 0.254 | 0.523 | 52.538 |
| DASR | 24.75 | **0.675** | 0.250 | 0.536 | 48.337 |
| LDM-30 | 24.49 | 0.651 | 0.248 | 0.572 | 50.895 |
| LDM-15 | **24.89** | 0.670 | 0.269 | 0.512 | 46.419 |
| ResShift-s15 | **24.90** | **0.673** | 0.228 | 0.603 | 53.897 |
| SinSR-s1 | 24.56 | 0.657 | **0.221** | **0.611** | 53.357 |
| *SinSR-1 +MGHF-n* | 24.31 | 0.645 | **0.225** | **0.660** | **55.323** |

Table 3: Quantitative comparison among widely used super-resolution models on *ImageNet-Test*. The best and second best results are highlighted in **red** and **blue**, respectively.

We systematically add the content-style consistency (CSC) and local information preservation (LIP) objectives to the MGHF-n framework while training on OSEDiff (Wu et al., 2024a). The effect of these objectives is depicted in Table 6.

**Comparison of MGHF with LPIPS and naive perceptual loss.** We compare the efficacy of the proposed MGHF-n and MGHF-c with VGG-based naive perceptual loss (Johson et al., 2016) and LPIPS (Zhang et al., 2018) on DIV2K-Val, RealSet65, and RealSR test sets. From Table 4, we can deduce that simple MGHF-n outperforms both VGG-based naive perceptual loss and LPIPS on these datasets when implemented in SinSR (Wang et al., 2024c). This comparison is performed using NIQE, MUSIQ, and CLIPIQA metrics across two real-world datasets and one synthetic dataset.

**MGHF's performance gain on different downscaling factors.** We investigated the robustness of MGHF across different downscaling factors by applying it to OSEDiff. We downscaled the test set DrealSR (Wei et al., 2020) by factors $2\times$, $4\times$, $8\times$ using Real-ESRGAN (Wang et al., 2021c). We found that MGHF yields superior performance compared to the original OSEDiff across almost every metric for $2\times$, $4\times$, $8\times$ upscaling factors, as demonstrated in Table 7.

**MGHF's performance gain under different degradation methods.** We further evaluate the robustness of MGHF on diverse degradation methods using the DRealSR dataset (Wei et al., 2020). Specifically, we adopt degradations generated by Real-ESRGAN (Wang et al., 2021c), NDR (Yao et al., 2024), BSRDM (Yue et al., 2022), and DASR (Wang et al., 2021a). Our results on Table 8 show that OSEDiff+MGHF consistently outperforms OSEDiff under degradations such as rain, haze, noise, ISP signal, and real-world conditions.

**Time and space complexity of each component of MGHF.** We analyze the time and space complexity of each component of the MGHF-c objective in Table 5. The results show that the DFE feature extractor is more computationally efficient than the VGG-16 feature extractor.

**Effect of hyperparameters on the adaptive weighted detail feature extractor (AWDFE).** We investigate the effect of the hyperparameters $\alpha$ and $\gamma$ (see Eq. (5)) on the feature maps shown in Sec. C.1. We found that $\gamma$ has a stronger influence on the feature maps than $\alpha$. Based on empirical observations, we set $\alpha = 1.1$ and $\gamma = \frac{1}{\sqrt{2}}$ in our experiments.

Table 4: Ablation study of the proposed MGHF-n and widely used perceptual losses.

| Datasets | Methods | NIQE↓ | MUSIQ↑ | CLIPIQA↑ |
|---|---|---|---|---|
| DIV2K-Val | SinSR-s1 (Wang et al., 2024c) | 6.02 | 62.82 | 0.6471 |
| | SinSR-1 + Perceptual Loss[†] | 5.97 | 61.94 | 0.6713 |
| | SinSR-1 + LPIPS[‡] | 6.06 | 62.95 | 0.6638 |
| | *SinSR-1 + **MGHF-n*** | **5.80** | **63.69** | **0.6822** |
| RealSet65 | SinSR-s1 (Wang et al., 2024c) | 5.98 | 62.17 | 0.7150 |
| | SinSR-1 + Perceptual Loss[†] | 5.63 | 62.64 | 0.7343 |
| | SinSR-1 + LPIPS[‡] | 5.84 | 63.70 | 0.7295 |
| | *SinSR-1 + **MGHF-n*** | **5.54** | **63.97** | **0.7405** |
| RealSR[*] | SinSR-s1 (Wang et al., 2024c) | 6.29 | 60.80 | 0.6122 |
| | SinSR-1 + Perceptual Loss[†] | 6.15 | 62.43 | 0.6670 |
| | SinSR-1 + LPIPS[‡] | 6.36 | 61.84 | 0.6580 |
| | *SinSR-1 + **MGHF-n*** | 6.02 | 62.85 | 0.6740 |

[*] RealSR is pre-processed following Wu et al. (2024a).
[†] VGG-based perceptual loss Johnson et al. (2016).
[‡] LPIPS loss Zhang et al. (2018).

| Objective | GFLOPs | Memory (MB) | Params (M) |
|---|---|---|---|
| LIP Loss | 21.402 | 819.96 | 0.139 |
| Gram Loss | 8.590 | 512.13 | 0.000 |
| Correlation Loss | 0.369 | 512.00 | 0.000 |
| AWDFE MSE Loss | 0.067 | 256.00 | 0.000 |
| DFE MSE Loss | 0.067 | 384.00 | 0.000 |
| Detail Feature Extractor (DFE) | 72.290 | 1.31 | 0.34 |
| **TOTAL** | **102.785** | **2499.89** | **0.479** |
| VGG Feature Extractor | 160.36 | 56.13 | 14.71 |

Table 5: Time and Space Complexity of the MGHF-c Algorithm.

Table 6: Ablation study of each objective contribution on MGHF-c while applying on OSEDiff.

| Method Name | PSNR ↑ | SSIM ↑ | LPIPS ↓ |
|---|---|---|---|
| OSEDiff (original) | 27.9200 | 0.7835 | 0.2968 |
| MGHF-naive | 28.4000 | 0.7980 | 0.2839 |
| Correlation loss | 28.5996 | 0.7923 | 0.2943 |
| CSC | 28.6432 | 0.7931 | 0.2845 |
| Only AWDFE | 28.5005 | 0.7971 | 0.2818 |
| MGHF-Naive+AWDFE MSE | 28.4040 | 0.7991 | 0.2737 |
| MGHF-naive+CSC | 28.7218 | 0.7956 | 0.2813 |
| LIP Only loss | 28.6826 | 0.8008 | 0.2793 |
| MGHF-c (MGHF-naive+LIP+CSC) | 28.8702 | 0.8057 | 0.2713 |

Table 7: Comparison of OSEDiff and OSEDiff+MGHF under different upscaling factors on DrealSR dataset.

| Method | PSNR ↑ | SSIM ↑ | LPIPS ↓ | DISTS ↓ | CLIPIQA ↑ | NIQE ↓ | MUSIQ ↑ | MANIQA ↑ | FID ↓ |
|---|---|---|---|---|---|---|---|---|---|
| **2× Downscale by Real-ESRGAN (SR upscaling factor: 2×)** | | | | | | | | | |
| Original OSEDiff | 27.1099 | 0.7621 | 0.3240 | 0.2301 | 0.6947 | 6.3130 | 65.1418 | 0.5831 | 140.6872 |
| OSEDiff+MGHF | 28.2440 | 0.8007 | 0.2815 | 0.2156 | 0.6601 | 6.7596 | 64.7557 | 0.5957 | 133.1088 |
| **4× Downscale by Real-ESRGAN (SR upscaling factor: 4×)** | | | | | | | | | |
| Original OSEDiff | 25.7130 | 0.7082 | 0.4219 | 0.2842 | 0.6184 | 6.6597 | 57.0138 | 0.5335 | 169.3852 |
| OSEDiff+MGHF | 26.4370 | 0.7405 | 0.3545 | 0.2577 | 0.6309 | 6.7118 | 62.9601 | 0.5743 | 163.3906 |
| **8× Downscale by Real-ESRGAN (SR upscaling factor: 8×)** | | | | | | | | | |
| Original OSEDiff | 24.0767 | 0.6839 | 0.6058 | 0.4113 | 0.4688 | 9.0937 | 33.8054 | 0.4193 | 248.6803 |
| OSEDiff+MGHF | 23.9231 | 0.6825 | 0.4926 | 0.3406 | 0.5526 | 6.9721 | 55.0441 | 0.5359 | 221.2790 |

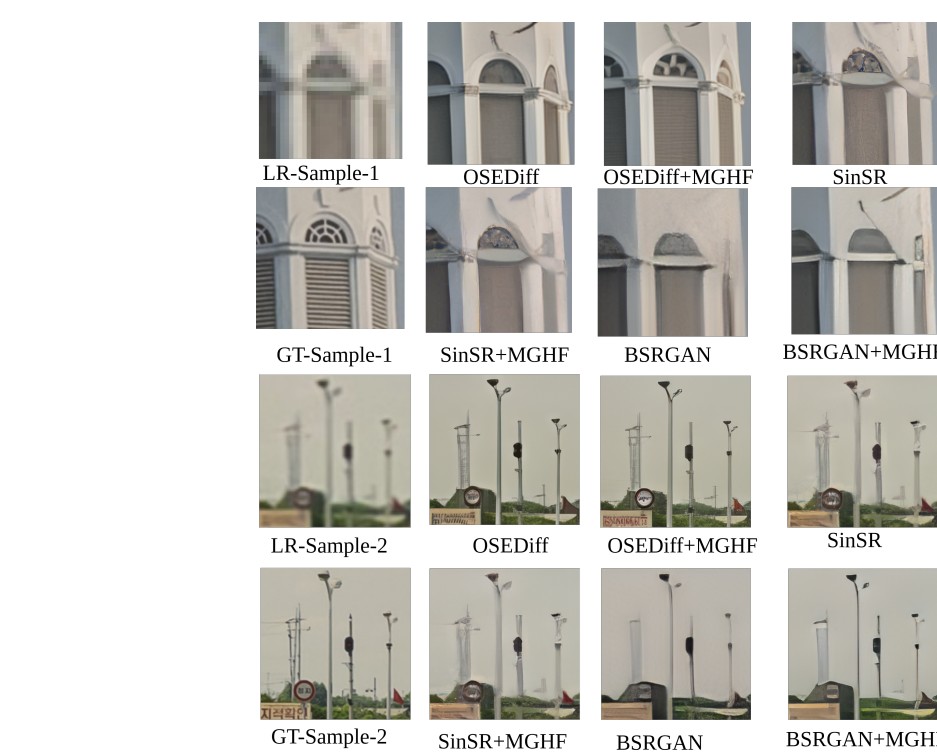

Figure 8: Qualitative comparisons of three state-of-the-art (SOTA) methods with and without the MGHF framework. Zoom in for a clearer view.

Table 8: Different Degradation Method Ablation Study on DrealSR dataset.

| Degradation | SR Method | PSNR ↑ | SSIM ↑ | LPIPS ↓ | DISTS ↓ | CLIPIQA ↑ | NIQE ↓ | MUSIQ ↑ | MANIQA ↑ | FID ↓ |
|---|---|---|---|---|---|---|---|---|---|---|
| Real-ESRGAN (Wang et al., 2021c) | OSEDiff | 25.7130 | 0.7082 | 0.4219 | 0.2842 | 0.6184 | 6.6597 | 57.0138 | 0.5335 | 169.3852 |
| | OSEDiff+MGHF | 26.4370 | 0.7405 | 0.3545 | 0.2577 | 0.6309 | 6.7118 | 62.9601 | 0.5743 | 163.3906 |
| NDR (Yao et al., 2024): Dehaze | OSEDiff | 29.3003 | 0.8293 | 0.2525 | 0.1973 | 0.7095 | 6.7410 | 66.9494 | 0.6238 | 108.4836 |
| | OSEDiff+MGHF | 30.6512 | 0.8431 | 0.2322 | 0.1940 | 0.6760 | 7.1101 | 66.4416 | 0.6264 | 104.8985 |
| NDR (Yao et al., 2024): Denoise | OSEDiff | 28.4467 | 0.8016 | 0.2884 | 0.2107 | 0.6894 | 6.3463 | 65.4359 | 0.5967 | 117.4478 |
| | OSEDiff+MGHF | 29.6504 | 0.8261 | 0.2517 | 0.2023 | 0.6781 | 6.6985 | 65.5262 | 0.6156 | 117.0953 |
| NDR (Yao et al., 2024): Derain | OSEDiff | 29.2999 | 0.8293 | 0.2525 | 0.1974 | 0.7093 | 6.7322 | 66.9490 | 0.6237 | 108.4542 |
| | OSEDiff+MGHF | 30.6535 | 0.8432 | 0.2321 | 0.1939 | 0.6764 | 7.1249 | 66.4288 | 0.6264 | 104.6391 |
| BSRDM (Yue et al., 2022): Gaussian (n=25) | OSEDiff | 23.5872 | 0.6722 | 0.3849 | 0.2503 | 0.6518 | 6.0765 | 62.2250 | 0.5426 | 146.5159 |
| | OSEDiff+MGHF | 23.9160 | 0.7023 | 0.3219 | 0.2311 | 0.6735 | 6.4936 | 64.2867 | 0.5988 | 136.2164 |
| BSRDM (Yue et al., 2022): JPEG | OSEDiff | 24.9658 | 0.7034 | 0.2974 | 0.2167 | 0.6864 | 6.2706 | 64.4699 | 0.5834 | 126.8771 |
| | OSEDiff+MGHF | 25.4904 | 0.7207 | 0.2700 | 0.2105 | 0.6576 | 7.0892 | 63.9669 | 0.6004 | 121.7394 |
| BSRDM (Yue et al., 2022): Signal | OSEDiff | 24.7141 | 0.7012 | 0.2883 | 0.2083 | 0.6948 | 6.6281 | 65.3582 | 0.6030 | 118.5541 |
| | OSEDiff+MGHF | 25.2224 | 0.7184 | 0.2637 | 0.2002 | 0.6723 | 6.9913 | 64.9218 | 0.6062 | 110.6171 |
| DASR (Wang et al., 2021a): Bicubic | OSEDiff | 29.2698 | 0.8219 | 0.2623 | 0.2029 | 0.7024 | 6.7434 | 66.2623 | 0.6138 | 115.8055 |
| | OSEDiff+MGHF | 30.5273 | 0.8401 | 0.2373 | 0.1966 | 0.6588 | 7.0826 | 65.3810 | 0.6122 | 110.6456 |
| DASR (Wang et al., 2021a): s-fold downsampler | OSEDiff | 27.4674 | 0.7835 | 0.2644 | 0.2032 | 0.6971 | 6.7787 | 66.1245 | 0.6121 | 115.7809 |
| | OSEDiff+MGHF | 28.1878 | 0.7954 | 0.2427 | 0.1980 | 0.6617 | 7.0105 | 65.5845 | 0.6146 | 112.7479 |

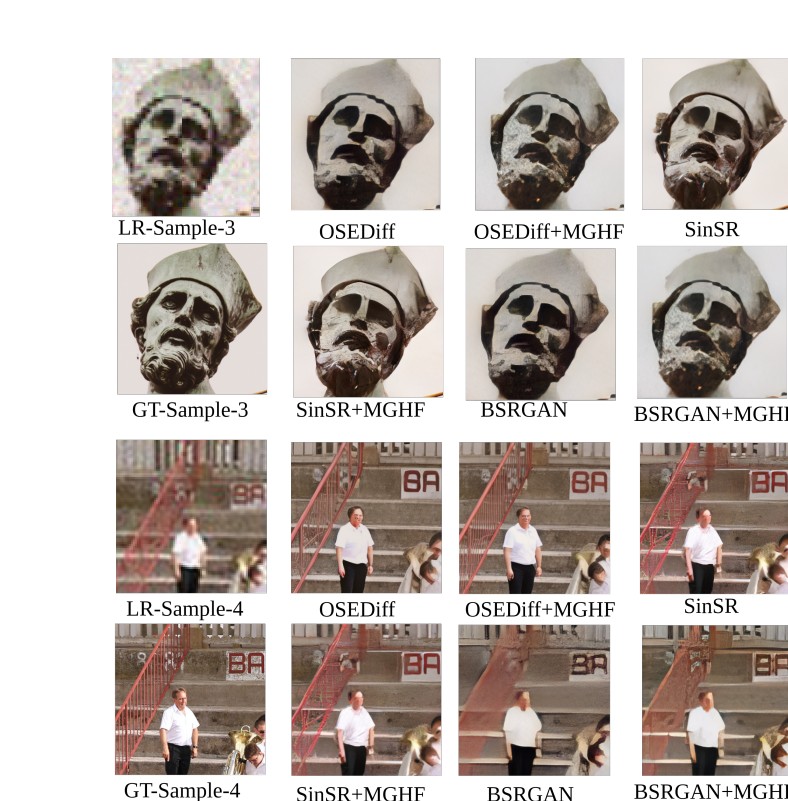

LR-Sample-3 OSEDiff OSEDiff+MGHF SinSR

GT-Sample-3 SinSR+MGHF BSRGAN BSRGAN+MGHF

LR-Sample-4 OSEDiff OSEDiff+MGHF SinSR

GT-Sample-4 SinSR+MGHF BSRGAN BSRGAN+MGHF

Figure 9: Qualitative comparisons of three state-of-the-art (SOTA) methods with and without the MGHF framework. Zoom in for a clearer view.

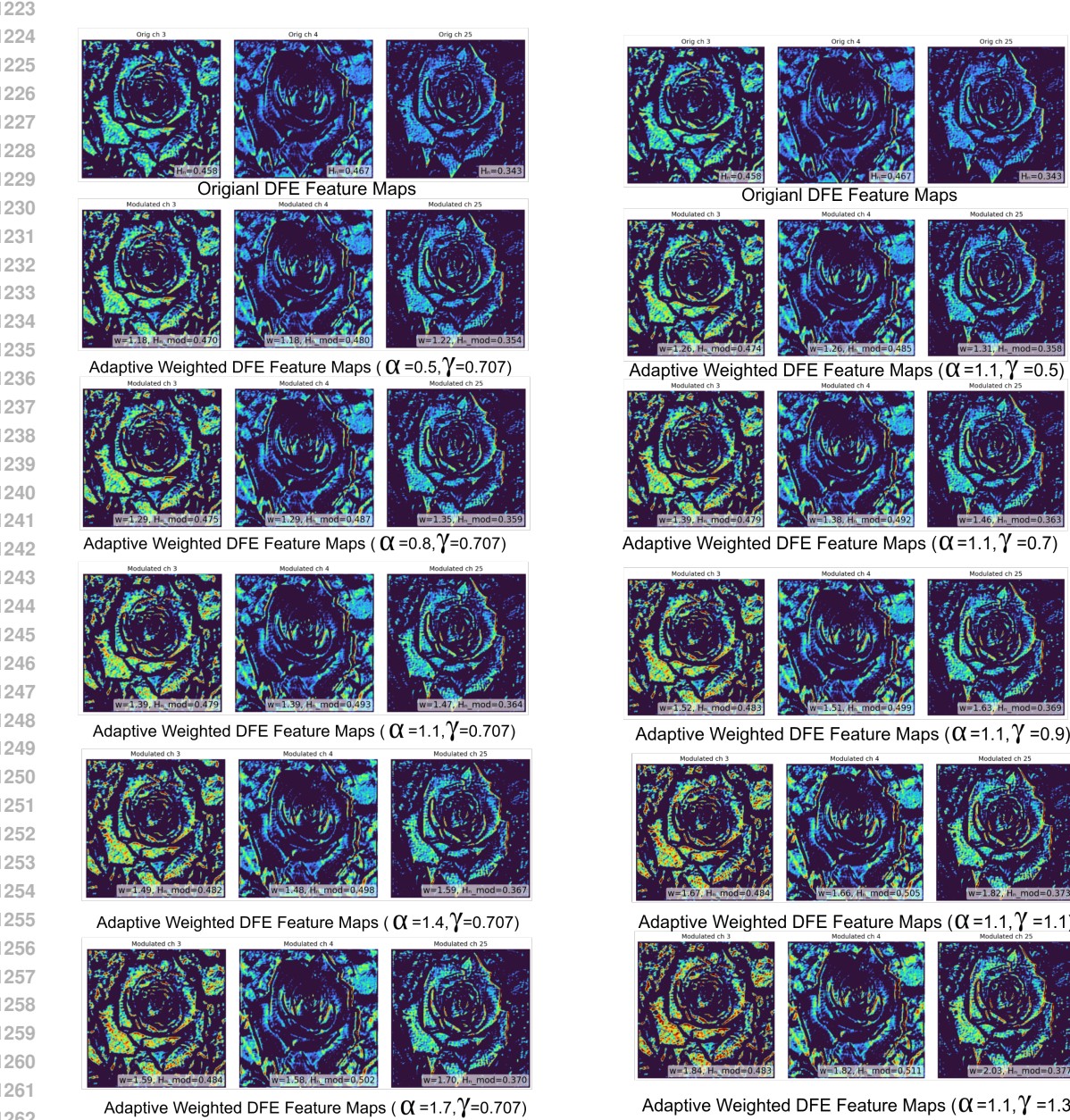

Effect of AWDFE feature maps by varying $\alpha$ and $\gamma$ remain constant.

Effect of AWDFE feature maps by varying $\gamma$ and $\alpha$ remain constant.

Figure 10: Effect of $\alpha$ and $\gamma$ in Eq. (5) on the adaptive weighted detail feature extractor.

Table 9: Comparison of OSEDiff and OSEDiff+MGHF on different datasets using QualiCLIP$^+$ and CLIPIQA metrics.

| Dataset | Method | QualiCLIP$^+$ ↑ | CLIPIQA ↑ |
|---------|--------|-----------------|-----------|
| DIV2K-Val | OSEDiff Original | 0.6689 | 0.6680 |
| | OSEDiff + MGHF-c | **0.6737** | **0.6735** |
| DRealSR | OSEDiff Original | 0.6566 | **0.6964** |
| | OSEDiff + MGHF-c | **0.6566** | 0.6955 |
| RealSR | OSEDiff Original | 0.6643 | **0.6686** |
| | OSEDiff + MGHF-c | **0.6672** | 0.6673 |

## C.2   PERFORMANCE COMPARISON BETWEEN CLIPIQA AND QUALICLIP$^+$ METRICS

CLIP-IQA (Wang et al., 2023) is a widely used image quality metric, yet it has notable limitations. The primary short-coming is its ability to only classify images as good or bad without providing explanations for its quality assessments. This limitation stems from a broader challenge inherent in off-the-shelf CLIP models: their focus on high-level semantics prevents them from generating quality-aware image representations, as they lack sensitivity to low-level image characteristics such as noise and blur. To address these limitations, QualiClip (Agnolucci et al., 2024) proposes a novel approach that trains CLIP to rank degraded images by measuring their similarity to quality-related antonym text prompts.

In our experiment ( Table 9), we observed that OSEDiff+MGHF outperforms OSEDiff in the QualiCLIP$^+$ metric across the DIV2K-Val (Agustsson and Timofte, 2017), DRealSR (Wei et al., 2020), and RealSR (Cai et al., 2019) datasets. However, when evaluated with the CLIP-IQA metric, OSEDiff+MGHF exhibits slightly lower or comparable performance to OSEDiff on the DRealSR and RealSR datasets. Furthermore, as shown in Table 1, MGHF improves performance across most metrics when integrated with OSEDiff on these three datasets.

## C.3   MORE PARAMETER DETAILS

In the detail feature extractor, before sending the image to the invertible neural network, we expand the image channels from 3 to $N$. In our experiment, we set $N = 128$. Moreover, in our experiment, we set the number of invertible blocks in the detail feature extractor to one. Finally, in the local information preservation objective, while calculating MoNCE (Zhan et al., 2022), we use $32 \times 32$ patches with a stride of 24 for selecting the neighboring patches.

## D  APPENDIX: PROOF

### D.1  PROPOSITION 1

**Proposition 1.** [Information Preservation] *The use of non-homeomorphic transform-based perceptual loss results in information approximation, whereas a diffeomorphic transform-based perceptual loss preserves all frequency components during translation. Consequently, the latter facilitates superior performance in perceptual loss calculation.*

*Proof.*
***Note***: *We will combine foundational concepts from functional analysis (Hilbert spaces, unitary operators (Schwinger, 1960)), measure theory (change of variables), and signal processing (Plancherel's theorem (Yoshizawa, 1954)) with a clear application to the decision-theoretic framework of machine learning for justifying this proposition.*

**[Diffeomorphic perceptual losses preserve frequency information]** Let $\Omega \subset \mathbb{R}^d$ be a bounded open set and let $\mathcal{X} = L^2(\Omega)$ with inner product $\langle f, g \rangle = \int_\Omega f(x)g(x)\,dx$. For a map $T : \mathcal{X} \to \mathcal{H}$ into a Hilbert space $\mathcal{H}$, define the perceptual loss

$$\mathcal{L}_T(f, g) \;=\; \|T(f) - T(g)\|_\mathcal{H}.$$

Then:

(i) If $T_{\mathrm{nh}}$ is non-homeomorphic (in particular, non-injective) on $\mathcal{X}$, there exist $f \neq g$ with $\mathcal{L}_T(f, g) = 0$. Thus the loss performs only an *information approximation*, collapsing some distinctions between inputs.

(ii) Let $\phi : \overline{\Omega} \to \overline{\Omega}$ be a $C^1$ diffeomorphism, and define the pullback operator

$$(U_\phi f)(y) \;=\; f\big(\phi^{-1}(y)\big) \sqrt{\big| \det D\phi^{-1}(y)\big|}.$$

Then $U_\phi$ is unitary on $L^2(\Omega)$ and

$$\mathcal{L}_U(f, g) = \|f - g\|_2.$$

Consequently, by Plancherel's theorem, the discrepancy energy across *all* Fourier frequencies is preserved; no frequency component is lost under $U_\phi$.

*(i) Non-homeomorphic case.* Since $T_{\mathrm{nh}}$ is not injective, by definition there exist $f \neq g$ in $\mathcal{X}$ with $T_{\mathrm{nh}}(f) = T_{\mathrm{nh}}(g)$. Hence, the perceptual loss is zero:

$$\mathcal{L}_T(f, g) = \|T_{\mathrm{nh}}(f) - T_{\mathrm{nh}}(g)\|_\mathcal{H} = 0.$$

However, since $f \neq g$, their $L^2$ distance is non-zero:

$$\|f - g\|_2 > 0.$$

This demonstrates that the loss metric $\mathcal{L}_T$ cannot distinguish between distinct signals $f$ and $g$, implying that information is necessarily discarded or approximated.

*(ii) Diffeomorphic case.* Let $\phi \in \mathrm{Diff}(\overline{\Omega})$ be a $C^1$ diffeomorphism. The operator $U_\phi$ is defined as the pullback, a generalization of the change of variables in integration. To prove that $U_\phi$ is unitary on $L^2(\Omega)$, we must show it preserves the inner product. For $f, g \in L^2(\Omega)$, we consider the inner product $\langle U_\phi f, U_\phi g \rangle$:

$$\langle U_\phi f, U_\phi g \rangle = \int_\Omega \big((U_\phi f)(y)\big)\,\big((U_\phi g)(y)\big)\,dy$$

Substituting the definition of $U_\phi$, we get:

$$= \int_\Omega f(\phi^{-1}(y)) \, g(\phi^{-1}(y)) \, \sqrt{\left| \det D\phi^{-1}(y) \right|}^2 \, dy$$

$$= \int_\Omega f(\phi^{-1}(y)) \, g(\phi^{-1}(y)) \, \left| \det D\phi^{-1}(y) \right| \, dy.$$

Now, we perform a change of variables using $x = \phi^{-1}(y)$. By the multi-variable change of variables formula, we have $dy = |\det D\phi(x)| \, dx$. Since $\phi$ is a diffeomorphism, $D\phi^{-1}(y) = (D\phi(x))^{-1}$ and thus $|\det D\phi^{-1}(y)| = |\det(D\phi(x))^{-1}| = |\det D\phi(x)|^{-1}$. Therefore, $dy = |\det D\phi(x)| \, dx = \frac{1}{|\det D\phi(y)|} \, dx$. Using the change of variables, the integral becomes:

$$\langle U_\phi f, U_\phi g \rangle = \int_{\phi(\Omega)} f(x) \, g(x) \, dx.$$

Since $\phi : \overline{\Omega} \to \overline{\Omega}$ is a diffeomorphism, it maps the domain $\Omega$ onto itself. Thus $\phi(\Omega) = \Omega$.

$$\langle U_\phi f, U_\phi g \rangle = \int_\Omega f(x) \, g(x) \, dx = \langle f, g \rangle.$$

This proves that $U_\phi$ is a unitary operator. A direct consequence of this is that the norm (and thus the distance) is preserved:

$$\mathcal{L}_U(f, g) = \|U_\phi f - U_\phi g\|_2 = \|f - g\|_2.$$

By Plancherel's theorem, which relates the energy of a signal to the energy of its Fourier transform, we have:

$$\|f - g\|_2^2 = \int_\mathbb{R} \left| \widehat{f}(\xi) - \widehat{g}(\xi) \right|^2 \, d\xi,$$

where $\widehat{f}(\xi)$ is the Fourier transform of $f$. Since $\mathcal{L}_U(f, g) = \|f - g\|_2$, the perceptual loss directly measures the total spectral energy of the difference between the signals. This means that no frequency component is ignored or annihilated by the transformation, thus preserving all frequency information.

**Conclusion for Part I:.** Diffeomorphic transformations, by their unitary nature, lead to a perceptual loss that is a perfect surrogate for the true $L^2$ distance, preserving all information including frequency components. Non-homeomorphic transformations, being non-injective, necessarily discard information.

---

**[Diffeomorphic perceptual losses are decision-theoretically superior]** Let $\Omega \subset \mathbb{R}^d$ be a bounded open set and $\mathcal{X} = L^2(\Omega)$ with inner product $\langle f, g \rangle = \int_\Omega f(x)g(x) \, dx$ and norm $\|f\|_2 = \sqrt{\langle f, f \rangle}$. For a (measurable) feature map $T : \mathcal{X} \to \mathcal{H}$ into a Hilbert space $\mathcal{H}$, define the *perceptual loss*

$$\mathcal{L}_T(f, g) := \|T(f) - T(g)\|_\mathcal{H}.$$

Given a distribution $P$ on pairs $(X, Y)$ with $Y \in \mathcal{X}$ and a hypothesis class $\mathcal{H}\dagger_{\sqrt{}}$ of predictors $h : \mathrm{dom}(X) \to \mathcal{X}$, define the population (expected) risks:

$$\mathcal{R}(h) := \mathbb{E}\big[\|h(X) - Y\|_2^2\big], \qquad \mathcal{R}_T(h) := \mathbb{E}\big[\|T(h(X)) - T(Y)\|_\mathcal{H}^2\big].$$

(a) (*Exact calibration*) Let $\phi : \overline{\Omega} \to \overline{\Omega}$ be a $C^1$ diffeomorphism and define the pullback $U_\phi$ as before. Then for all $h$, $\mathcal{R}_U(h) = \mathcal{R}(h)$. Consequently, for every hypothesis class $\mathcal{H}\dagger_{\sqrt{}}$, $\arg\min_{h \in \mathcal{H}\dagger_{\sqrt{}}} \mathcal{R}_U(h) = \arg\min_{h \in \mathcal{H}\dagger_{\sqrt{}}} \mathcal{R}(h)$.

(b) (*Strict suboptimality of non-homeomorphic transforms*) If $T$ is non-injective, there exist a distribution $P$ and a hypothesis class $\mathcal{H}^\dagger_{\sqrt{}}$ for which any minimizer of $\mathcal{R}_T$ has strictly larger true risk than a minimizer of $\mathcal{R}$.

*(a) Exact calibration.* As shown in the previous section, the operator $U_\phi$ is unitary on $L^2(\Omega)$, which implies it is an isometry, preserving distances: $\|U_\phi f - U_\phi g\|_2 = \|f - g\|_2$. Applying this property to the risk functions, we have:

$$\mathcal{R}_U(h) = \mathbb{E}\big[\|U_\phi h(X) - U_\phi Y\|_2^2\big]$$

Since the norm is preserved, this simplifies to:

$$\mathcal{R}_U(h) = \mathbb{E}\big[\|h(X) - Y\|_2^2\big] = \mathcal{R}(h).$$

This shows the risks are identical for any predictor $h$. Therefore, the set of minimizers for the perceptual risk is exactly the same as the set of minimizers for the true risk:

$$\arg \min_{h \in \mathcal{H}^\dagger_{\sqrt{}}} \mathcal{R}_U(h) = \arg \min_{h \in \mathcal{H}^\dagger_{\sqrt{}}} \mathcal{R}(h).$$

This property is crucial as it guarantees that a model trained to minimize the perceptual loss will also be optimal with respect to the true objective.

*(b) Strict suboptimality.* Because $T$ is non-injective, there exist at least two distinct signals $u, v \in \mathcal{X}$ such that $u \neq v$ but $T(u) = T(v)$. Let's set a specific value for this collapsed point, $T(u) = T(v) =: z$. We construct a counterexample. Let the distribution $P$ be a simple Bernoulli distribution on the pair $(X, Y)$, where $X$ is arbitrary and

$$Y = \begin{cases} u, & \text{with probability } 1/2, \\ v, & \text{with probability } 1/2. \end{cases}$$

Now, let the hypothesis class $\mathcal{H}^\dagger_{\sqrt{}}$ be the set of constant predictors, $h_w(\cdot) \equiv w$, for any $w \in \mathcal{X}$.

First, consider the perceptual risk $\mathcal{R}_T(h_w)$:

$$\mathcal{R}_T(h_w) = \mathbb{E}\big[\|T(h_w(X)) - T(Y)\|_{\mathcal{H}}^2\big]$$

$$= \tfrac{1}{2}\|T(w) - T(u)\|_{\mathcal{H}}^2 + \tfrac{1}{2}\|T(w) - T(v)\|_{\mathcal{H}}^2.$$

Since $T(u) = T(v) = z$, this simplifies to:

$$\mathcal{R}_T(h_w) = \tfrac{1}{2}\|T(w) - z\|_{\mathcal{H}}^2 + \tfrac{1}{2}\|T(w) - z\|_{\mathcal{H}}^2 = \|T(w) - z\|_{\mathcal{H}}^2.$$

This risk is minimized when $T(w) = z$. Therefore, any constant predictor $h_w$ where $T(w) = z$ is a minimizer of the perceptual risk. This includes $h_u$ and $h_v$.

Next, consider the true risk $\mathcal{R}(h_w)$:

$$\mathcal{R}(h_w) = \mathbb{E}\big[\|h_w(X) - Y\|_2^2\big] = \tfrac{1}{2}\|w - u\|_2^2 + \tfrac{1}{2}\|w - v\|_2^2.$$

We want to find the true risk of the minimizers of the perceptual loss. The perceptual minimizers are $h_u$ and $h_v$. The true risk for the minimizer $h_u$ is:

$$\mathcal{R}(h_u) = \tfrac{1}{2}\|u - u\|_2^2 + \tfrac{1}{2}\|u - v\|_2^2 = \tfrac{1}{2}\|u - v\|_2^2.$$

The true risk for the minimizer $h_v$ is:

$$\mathcal{R}(h_v) = \tfrac{1}{2}\|v - u\|_2^2 + \tfrac{1}{2}\|v - v\|_2^2 = \tfrac{1}{2}\|u - v\|_2^2.$$

Now, let's find the true minimizer of $\mathcal{R}(h_w)$. To minimize $\frac{1}{2}\|w - u\|_2^2 + \frac{1}{2}\|w - v\|_2^2$, we can take the derivative with respect to $w$ and set it to zero, which gives the true optimal constant predictor as the mean $w^\star = \frac{u+v}{2}$. The minimum true risk is:

$$\mathcal{R}(h_w) = \frac{1}{2}\left\|\frac{u+v}{2} - u\right\|_2^2 + \frac{1}{2}\left\|\frac{u+v}{2} - v\right\|_2^2 = \frac{1}{2}\left\|\frac{v-u}{2}\right\|_2^2 + \frac{1}{2}\left\|\frac{u-v}{2}\right\|_2^2$$

$$= \frac{1}{2} \cdot \frac{1}{4}\|u - v\|_2^2 + \frac{1}{2} \cdot \frac{1}{4}\|u - v\|_2^2 = \frac{1}{4}\|u - v\|_2^2.$$

We can compare the true risk of the perceptual minimizer $h_u$ with the true optimal risk:

$$\mathcal{R}(h_u) = \frac{1}{2}\|u - v\|_2^2 = \frac{1}{4}\|u - v\|_2^2 + \frac{1}{4}\|u - v\|_2^2 = \mathcal{R}(h_w) + \frac{1}{4}\|u - v\|_2^2.$$

Since $u \neq v$, we have $\|u - v\|_2^2 > 0$, and thus the gap is strictly positive. This shows that the solution found by minimizing the non-homeomorphic perceptual loss is strictly suboptimal for the true objective.

**Summary.** This proof provides that:

1. **Diffeomorphic losses:** The unitary nature of the transformation ensures the perceptual loss is exactly equal to the true $L^2$ loss, for any signals, distributions, and hypothesis classes. This provides a guarantee that the model learns the correct underlying objective.

2. **Non-homeomorphic losses:** Their non-injective nature means they can map distinct signals to the same output. A model trained with such a loss can be "tricked" into finding a solution that appears optimal for the loss function but is demonstrably and strictly suboptimal for the true objective, leading to a poorer generalization or performance.

Therefore, a diffeomorphic transform-based invertible neural network is a theoretically superior choice than a non-homomorphic transform-based CNN for perceptual losses.

□

## D.2 TOY EXAMPLE OF INFORMATION PRESERVATION BY DIFFEOMOPHIC TRANSFORM

## D.3 SWIRLED TRANSFORM

We define a swirled transformation that applies radially-dependent rotation around the center $(x_0, y_0)$. Given pixel coordinates $(x, y)$, we compute centered coordinates and polar representation, then apply quadratic angular displacement proportional to distance-squared. This invertible transform preserves topological structure while introducing controlled spiral distortion for data augmentation.

$$x_c = x - x_0, \qquad y_c = y - y_0,$$
$$r = \sqrt{x_c^2 + y_c^2}, \qquad \theta = \text{atan2}(y_c, x_c),$$
$$\theta' = \theta + k\, r^2,$$
$$x' = x_0 + r\cos\theta', \qquad y' = y_0 + r\sin\theta'.$$

## D.4 COLLAPSED TRANSFORM

We introduce non-invertible collapse transformations for dimensionality reduction and feature compression. The radial variant contracts points toward center $(x_0, y_0)$ via scaling function $g(r) \in [0, 1]$, enabling controllable information loss. Setting $g(r) = 0$ yields complete collapse, while $g(r) < 1$ provides partial compression. Orthogonal projection represents the simplest linear collapse operation.

$$x' = x_0 + g(r)\,(x - x_0),$$
$$y' = y_0 + g(r)\,(y - y_0),$$
$$\text{where } r = \sqrt{(x - x_0)^2 + (y - y_0)^2}, \quad g : [0, \infty) \to [0, 1].$$

Simple collapse transform onto the x-axis:

$$(x', y') = (x,\, 0).$$

In Fig. 11, the diffeomorphic swirled transform preserves high-frequency noise during loss computation, whereas the non-homomorphic collapsed transform fails to do so.

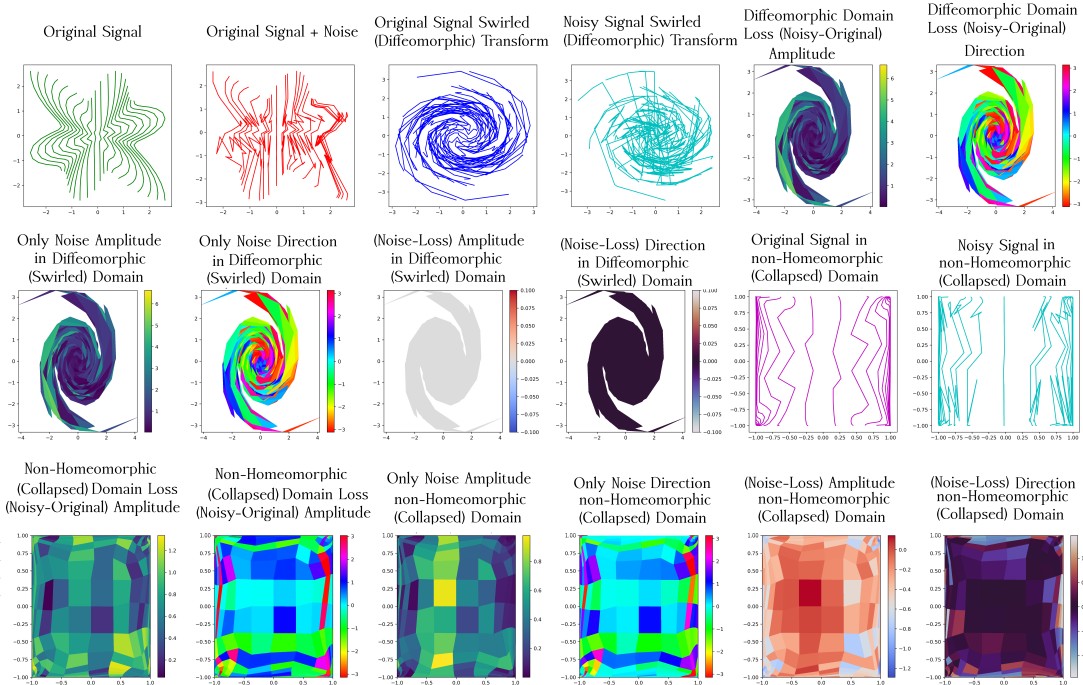

Figure 11: Toy examples illustrating high-frequency preservation in diffeomorphic transforms, whereas non-homomorphic transforms exhibit information loss.

## D.5 COROLLARY 1.

**Corollary 1.** [Frequency distortion by ReLU operation] *The output signal $y(t) = ReLU(\cos(\omega_0 t))$ contains frequency components at integer multiples of $\omega_0$ that were not present in the input signal $x(t) = \cos(\omega_0 t)$.*

*Proof.* Let $y(t) = \text{ReLU}(\cos(\omega_0 t))$. The complex Fourier series coefficients $c_n$ for $y(t)$ are given by:

$$c_0 = \frac{1}{\pi}$$

$$c_1 = c_{-1} = \frac{1}{4}$$

$$c_n = \frac{\cos(n\pi/2)}{\pi(1 - n^2)} \quad \text{for } |n| > 1$$

The signal $y(t)$ is periodic with period $T_0 = 2\pi/\omega_0$. The Fourier coefficients $c_n$ are calculated by the integral:

$$c_n = \frac{1}{T_0} \int_0^T y(t) e^{-jn\omega t} dt$$

The function $\cos(\omega_0 t)$ is positive on the interval $[-T_0/4, T_0/4]$ within one period. Therefore, the integral simplifies to:

$$c_n = \frac{1}{T_0} \int_{-T/4}^{T/4} \cos(\omega_0 t) e^{-jn\omega t} dt$$

Using Euler's formula, $\cos(\theta) = \frac{1}{2}(e^{j\theta} + e^{-j\theta})$, we get:

$$c_n = \frac{1}{2T_0} \int_{-T/4}^{T/4} (e^{j\omega t} + e^{-j\omega t}) e^{-jn\omega t} dt$$

$$= \frac{1}{2T_0} \int_{-T/4}^{T/4} \left( e^{j(1-n)\omega t} + e^{-j(1+n)\omega t} \right) dt$$

For the case where $n \neq \pm 1$, we can integrate directly:

$$
\begin{aligned}
c_n &= \frac{1}{2T_0} \left[ \frac{e^{j(1-n)\omega t}}{j(1-n)\omega_0} - \frac{e^{-j(1+n)\omega t}}{j(1+n)\omega_0} \right]_{-T/4}^{T/4} \\
&= \frac{1}{2T_0} \left( \frac{2\sin((1-n)\pi/2)}{(1-n)\omega_0} + \frac{2\sin((1+n)\pi/2)}{(1+n)\omega_0} \right) \\
&= \frac{1}{2\pi(1-n^2)} \left( (1+n)\sin(\pi/2 - n\pi/2) + (1-n)\sin(\pi/2 + n\pi/2) \right) \qquad (19) \\
&= \frac{1}{2\pi(1-n^2)} \left( (1+n)\cos(n\pi/2) + (1-n)\cos(n\pi/2) \right) \\
&= \frac{2\cos(n\pi/2)}{2\pi(1-n^2)} = \frac{\cos(n\pi/2)}{\pi(1-n^2)}
\end{aligned}
$$

The special cases for $n = 0$ and $n = \pm 1$ must be calculated separately, yielding $c_0 = 1/\pi$ and $c_{\pm 1} = 1/4$.

The output signal $y(t) = \text{ReLU}(\cos(\omega_0 t))$ contains frequency components at integer multiples of $\omega_0$ that were not present in the input signal $x(t) = \cos(\omega_0 t)$.

The input signal $x(t)$ is band-limited, containing only frequencies at $\pm\omega_0$. From Eq. (19), we can evaluate the coefficients $c_n$ for $|n| > 1$. For example, for the second harmonic ($n = 2$):

$$c_2 = \frac{\cos(\pi)}{\pi(1-4)} = \frac{-1}{-3\pi} = \frac{1}{3\pi} \neq 0$$

And for the fourth harmonic ($n = 4$):

$$c_4 = \frac{\cos(2\pi)}{\pi(1-16)} = \frac{1}{-15\pi} \neq 0$$

Since $c_n$ is non-zero for even integers $n \geq 2$, the Fourier series representation of $y(t)$ contains terms for frequencies $2\omega_0, 4\omega_0, \ldots$. These are new high-frequency components.

Applying the ReLU activation function to a band-limited signal can produce an output signal that is not band-limited to the original frequency range.

Let $x(t) = \cos(\omega_0 t)$ be a signal band-limited to the frequency $\omega_0$. Its Fourier Transform contains energy only at $\omega = \pm\omega_0$. Let $y(t) = \text{ReLU}(x(t))$. By Eq. (19), the Fourier series of $y(t)$ contains non-zero coefficients corresponding to frequencies $n\omega_0$ for even integers $n \geq 2$. The existence of these harmonics implies that the Fourier Transform of $y(t)$ is non-zero for frequencies $|\omega| > \omega_0$. Therefore, the output signal $y(t)$ is no longer band-limited to the original frequency $\omega_0$, proving that the ReLU function has introduced new higher-frequency components. $\square$

## D.6 THEOREM 1

**Theorem 1.** [Superiority of INN over CNN in perceptual loss calculation]
*Invertible Neural Networks (INNs) offer theoretical advantages over Convolutional Neural Networks (CNNs) when used as perceptual feature extractors. Formally, let $f : \mathbb{R}^n \to \mathbb{R}^n$ denote a diffeomorphic INN and $g : \mathbb{R}^n \to \mathbb{R}^m$ a standard CNN feature map with non-invertible operators (pooling, ReLU, strided convolutions). Then, the following contrasts hold:*

- *Information conservation. INN: $H(f(X)) = H(X)$ (entropy preserved due to bijectivity). CNN: $H(g(X)) < H(X)$ (irreversible compression due to non-invertibility).*
- *Manifold preservation. INN: diffeomorphic mappings preserve topology of the image manifold. CNN: distortion mappings collapse neighborhoods and destroy manifold structure.*
- *Statistical equivalence. INN: all statistical moments of $X$ are preserved in $f(X)$. CNN: higher-order moments are altered or lost.*
- *Spectral completeness. INN: full frequency spectrum preserved, including high-frequency details. CNN: effective low-pass filtering due to pooling and convolution kernels.*
- *Gradient stability. INN: Jacobians are well-conditioned ($\det J_f(x) \neq 0$). CNN: singular Jacobians induce unstable or vanishing gradients.*
- *Distribution matching. INNs theoretically achieve perfect distribution matching, whereas CNNs exhibit positive Wasserstein distance.*

*Proof.* D.6.1 INFORMATION CONSERVATION

**[CNN Information Destruction]**

Any CNN with non-invertible operations (pooling, ReLU) necessarily destroys information. Specifically, for CNN function $g : \mathbb{R}^n \to \mathbb{R}^m$:

$$H(X) > H(g(X)) \tag{20}$$

where $H(\cdot)$ denotes differential entropy.

Consider max pooling operation $Pooling : \mathbb{R}^4 \to \mathbb{R}$ defined as $Poolin(x_1, x_2, x_3, x_4) = \max\{x_1, x_2, x_3, x_4\}$.

The mapping is not injective since multiple inputs map to the same output. For example, $(4, 1, 2, 3)$ and $(4, 0, 1, 2)$ both map to $4$.

By the data processing inequality:

$$I(X; Pooling(X)) \leq I(X; X) = H(X) \tag{21}$$

Since $P$ is not invertible, the inequality is strict: $I(X; Pooling(X)) < H(X)$.

For ReLU activation $\sigma(x) = \max(0, x)$, the function maps all negative values to zero, creating information loss quantified by:

$$H(X) - H(\sigma(X)) = \int_{-\infty}^{0} p_X(x) \log p_X(x) \, dx > 0 \tag{22}$$

where $p_X$ is the probability density of $X$.

**[Information Preservation in INNs]**
For any invertible neural network $f$ and random variable $X$:

$$H(f(X)) = H(X) \tag{23}$$

Since $f$ is invertible with inverse $f^{-1}$, we have:

$$H(f(X)) = -\int p_{f(X)}(y) \log p_{f(X)}(y) \, dy \tag{24}$$

$$= -\int p_X(f^{-1}(y)) \left| \det\left(\frac{\partial f^{-1}}{\partial y}\right) \right| \log\left( p_X(f^{-1}(y)) \left| \det\left(\frac{\partial f^{-1}}{\partial y}\right) \right| \right) dy \tag{25}$$

Using the change of variables $x = f^{-1}(y)$, $dx = \left| \det\left(\frac{\partial f}{\partial y}\right) \right| dy$:

$$H(f(X)) = -\int p_X(x) \log\left( p_X(x) \left| \det\left(\frac{\partial f^{-1}}{\partial f(x)}\right) \right| \right) dx \tag{26}$$

$$= -\int p_X(x) \log p_X(x) \, dx - \int p_X(x) \log\left| \det\left(\frac{\partial f^{-1}}{\partial f(x)}\right) \right| dx \tag{27}$$

$$= H(X) - \mathbb{E}_X\left[ \log\left| \det\left(\frac{\partial f}{\partial x}\right)^{-1} \right| \right] \tag{28}$$

$$= H(X) + \mathbb{E}_X\left[ \log\left| \det\left(\frac{\partial f}{\partial x}\right) \right| \right] \tag{29}$$

For coupling layers in INNs, the Jacobian determinant is designed to have unit absolute value, making the expectation zero, thus $H(f(X)) = H(X)$.

### D.6.2 Manifold Preservation Theory

[Natural Image Manifold Preservation] Let $M \subset \mathbb{R}^n$ be the natural image manifold. INNs preserve manifold structure while CNNs create distortions.

For INN $f : M \to M$, since $f$ is bijective and differentiable:

1. $f$ is a homeomorphism preserving topological structure

2. The tangent space structure is preserved: $T_{f(x)}M = df_x(T_xM)$

3. Geodesic distances are preserved up to the Riemannian metric transformation

For CNN $g : M \to M'$ where $\dim(M') < \dim(M)$ due to information loss:

$$\exists x_1, x_2 \in M : x_1 \neq x_2 \text{ but } g(x_1) = g(x_2) \tag{30}$$

This violates injectivity and creates manifold collapse, fundamentally distorting the natural image structure.

### D.6.3 STATISTICAL DISTRIBUTION THEORY

**[Moment Preservation in INNs]** For invertible function $f$ and random variable $X$:

$$\mathbb{E}[X^k] = \mathbb{E}[(f^{-1}(f(X)))^k] = \mathbb{E}[X^k], \quad \forall k \in N \tag{31}$$

Since $f$ is invertible, $f^{-1}(f(X)) = X$ almost surely. Therefore:

$$\mathbb{E}[(f^{-1}(f(X)))^k] = \mathbb{E}[X^k] \tag{32}$$

This preservation extends to all statistical moments, ensuring complete distributional equivalence.

**[CNN Moment Distortion]** For CNN with information-destroying operations, higher-order moments are not preserved:

$$\mathbb{E}[X^k] \neq \mathbb{E}[(g^\dagger(g(X)))^k] \text{ for } k \geq 2 \tag{33}$$

where $g^\dagger$ represents the pseudo-inverse reconstruction.

**Perceptual Loss Optimality** [INN Perceptual Loss Optimality] INN-based perceptual loss achieves theoretical minimum distortion:

$$D_{\text{INN}}^* = \inf_{f \in \mathcal{F}_{\text{INN}}} \mathbb{E}[\|X - f^{-1}(f(X))\|_2^2] = 0 \tag{34}$$

For perfect invertible reconstruction:

$$D_{\text{INN}}^* = \mathbb{E}[\|X - f^{-1}(f(X))\|_2^2] \tag{35}$$
$$= \mathbb{E}[\|X - X\|_2^2] \tag{36}$$
$$= 0 \tag{37}$$

In contrast, for CNNs with pseudo-inverse $g^\dagger$:

$$D_{\text{CNN}} = \mathbb{E}[\|X - g^\dagger(g(X))\|_2^2] > 0 \tag{38}$$

This limitation arises from the information loss of CNNs, as shown in Sec. D.6.1.

### D.6.4 SPECTRAL PROPERTIES OF CNNS VS. INNS

The difference in how CNNs and INNs handle frequency information stems from their core mathematical designs: CNNs use non-bijective operations, while INNs rely on bijective transformations.

**CNN: Low-Pass Filtering**
The primary culprit for a CNN's low-pass filtering behavior is the **pooling layer**, which performs non-invertible downsampling.

Let's consider a simple $2 \times 2$ average pooling operation on a discrete signal $x[n, m]$. The output signal $y[n, m]$ is given by:

$$y[n, m] = \frac{1}{4} \sum_{i=0}^{1} \sum_{j=0}^{1} x[2n + i, 2m + j]$$

This operation discards information. In the frequency domain, this downsampling without an anti-aliasing filter causes high-frequency content to alias into the low-frequency spectrum. The new spectrum is a superposition of the original spectrum and its shifted, aliased versions. Since this is a many-to-one mapping, the original high-frequency details cannot be recovered, leading to a permanent loss of information. This fundamentally proves the low-pass filtering effect.

**INN: Spectral Completeness**
INNs, by design, are composed of layers that perform bijective transformations. The key mathematical property is that the Jacobian determinant for each layer is non-singular (i.e., non-zero).

Let $f : \mathbb{R}^D \to \mathbb{R}^D$ be a layer in an INN. Its Jacobian matrix is $J_f(x) = \frac{\partial f(x)}{\partial x}$. For the mapping to be invertible, the determinant of this matrix must be non-zero for all inputs $x$:

$$|\det(J_f(x))| \neq 0$$

For a complete INN, which is a composition of $N$ such layers, $f_{\text{total}} = f_N \circ \cdots \circ f_1$, the overall Jacobian determinant is the product of the individual layers' determinants:

$$|\det(J_{f_{\text{total}}}(x))| = \prod_{i=1}^{N} |\det(J_f(x))| \neq 0$$

This non-zero determinant ensures that the transformation is a **diffeomorphism** and that a unique inverse exists. This means no information, including high-frequency content, is ever collapsed or destroyed. The original signal can be perfectly reconstructed from the output, thus proving the spectral preservation of INNs.

### D.6.5 GRADIENT FLOW STABILITY

[INN Gradient Preservation] INNs maintain gradient structure while CNNs suffer degradation:

$$\nabla_x L = J_f^T \nabla_{f(x)} L \tag{39}$$

where $J_f$ is the Jacobian of the INN transformation $f$.

For invertible $f$ with well-conditioned Jacobian:

$$\sigma_{\min}(J_f) \geq \epsilon > 0 \tag{40}$$

where, $\sigma_{\min}$ is the minimum singular value of a matrix. It's a scalar value that comes from the Singular Value Decomposition (SVD). The gradient transformation preserves magnitude:

$$\sigma_{\min}(J_f) \left\| \nabla_{f(x)} L \right\|_2 \leq \left\| J_f^T \nabla_{f(x)} L \right\|_2 \leq \sigma_{\max}(J_f) \left\| \nabla_{f(x)} L \right\|_2 \tag{41}$$

For CNNs with potentially singular Jacobian $J_g$ due to information loss:

$$\left\| J_g^T \nabla_{g(x)} L \right\|_2 \leq \sigma_{\max}(J_g) \left\| \nabla_{g(x)} L \right\|_2 \tag{42}$$

with $\sigma_{\min}(J_g) \to 0$ causing gradient vanishing in certain directions.

### D.6.6 WASSERSTEIN DISTANCE ANALYSIS

[Distribution Matching Optimality] INNs achieve perfect distribution matching while CNNs exhibit positive Wasserstein distance:

$$W_1\big(P_X, P_{f^{-1}(f(X))}\big) = 0 \quad \text{(INN)} \tag{43}$$

$$W_1\big(P_X, P_{g^\dagger(g(X))}\big) > 0 \quad \text{(CNN)} \tag{44}$$

For INNs, since $f^{-1}(f(X)) = X$ almost surely:

$$W_1\Big(P_X, P_{f^{-1}(f(X))}\Big) = W_1(P_X, P_X) = 0. \tag{45}$$

For CNNs with information loss, the distributions differ:

$$W_1(P_X, P_{g^\dagger(g(X))}) = \int |F_X(t) - F_{g^\dagger(g(X))}(t)|dt > 0 \tag{46}$$

$$W_1\Big(P_X, P_{f^{-1}(f(X))}\Big) = W_1(P_X, P_X) = 0 \tag{47}$$

where $F$ denotes cumulative distribution functions.

$\square$

## D.7 THEORY 2

**Theorem 2.** The $\mathcal{L}_{LIP}$ objective provides a tighter lower bound on mutual information than standard InfoNCE.

$$I(G; S) \geq \log N_k - \mathcal{L}_{LIP} \geq \log N_k - \mathcal{L}_{\text{InfoNCE}} \tag{48}$$

*Proof.* **Step 1: Setup.** For one feature map $k$ with $N_k$ patches, define the modulation

$$m(s_{ki}, g_{kj}) = \begin{cases} 1, & j = i, \\ Q(N_k - 1)a_{ij}^k, & j \neq i, \end{cases} \tag{49}$$

where $a_{ij}^k \in [0, 1]$ are Sinkhorn OT weights and $Q$ is a scalar. Define

$$r(s, g) = m(s, g) \exp\big(\tfrac{1}{\tau} s^\top g\big). \tag{50}$$

The per-query loss is

$$L_{i,k} = -\mathbb{E}\Big[\log r(s_{ki}, g_{ki}) - \log \sum_{j=1}^N r(s_{ki}, g_{kj})\Big]. \tag{51}$$

**Step 2: AM–GM bound on the log-sum.** By the arithmetic–geometric mean inequality,

$$\sum_{j=1}^N r(s, g_j) \geq N_k \Big(\prod_{j=1}^N r(s, g_j)\Big)^{1/N}. \tag{52}$$

Taking logs,

$$\log \sum_{j=1}^{N} r(s, g_j) \geq \log N_k + \frac{1}{N_k} \sum_{j=1}^{N} \log r(s, g_j). \tag{53}$$

Therefore,

$$\mathbb{E}\left[\log \sum_{j=1}^{N} r(s_{ki}, g_{kj})\right] \geq \log N_k + \frac{1}{N_k} \sum_{j=1}^{N} \mathbb{E}[\log r(s_{ki}, g_{kj})]. \tag{54}$$

**Step 3: Lower bound on** $\log N_k - L_{i,k}$. Substituting into the definition of $L_{i,k}$,

$$\log N_k - L_{i,k} = \log N_k + \mathbb{E}[\log r(s_{ki}, g_{ki})] - \mathbb{E}\left[\log \sum_{j} r(s_{ki}, g_{kj})\right]$$

$$\geq \mathbb{E}[\log r(s_{ki}, g_{ki})] - \frac{1}{N_k} \sum_{j=1}^{N} \mathbb{E}[\log r(s_{ki}, g_{kj})]. \tag{55}$$

If negatives $\{g_{kj}\}_{j \neq i}$ are i.i.d. from the marginal $p(g)$, then

$$\log N_k - L_{i,k} \geq \mathbb{E}[\log r(s_{ki}, g_{ki})] - \mathbb{E}_s\left[\mathbb{E}_{g \sim p(g)}[\log r(s_{ki}, g)]\right]. \tag{56}$$

**Step 4: Split into InfoNCE part and a gain term.** Expanding $r(s, g)$,

$$\log N_k - L_{i,k} \geq \left(\mathbb{E}\left[\frac{1}{\tau} s_{ki}^{\top} g_{ki}\right] - \mathbb{E}_s\left[\mathbb{E}_g\left[\frac{1}{\tau} s_{ki}^{\top} g\right]\right]\right)$$

$$+ \left(\mathbb{E}[\log m(s_{ki}, g_{ki})] - \mathbb{E}_s\left[\mathbb{E}_g[\log m(s_{ki}, g)]\right]\right). \tag{57}$$

The first parenthesis is exactly the InfoNCE lower bound. Denote the second parenthesis by $\Delta$. Thus

$$\log N_k - L_{i,k} \geq \left(\log N_k - L_{i,k}^{\text{InfoNCE}}\right) + \Delta. \tag{58}$$

**Step 5: Evaluate $\Delta$ for LIP.** For LIP, $m(s, g) = 1$ if $g$ is the positive pair and $m(s, g) = Q(N_k - 1)a_{ij}^k$ otherwise. Hence

$$\Delta = -\mathbb{E}\left[\log\left(Q(N_k - 1)a_{ij}^k\right)\right]. \tag{59}$$

Choosing $Q = \frac{1}{N-1}$ gives $\Delta = -\mathbb{E}[\log a_{ij}^k] \geq 0$, with strict inequality whenever $\mathbb{P}(a_{ij}^k < 1) > 0$.

**Step 6: Conclude.** Averaging over all patches and feature maps,

$$I(G; S) \geq \log N_k - \mathcal{L}_{\text{LIP}} \geq \log N_k - \mathcal{L}_{\text{InfoNCE}}, \tag{60}$$

with an additional non-negative gap $\Delta$. Therefore, LIP provides a strictly tighter lower bound than InfoNCE. □

