# OpenReview forum: "MGHF: Multi-Granular High-Frequency Perceptual Loss for Image Super-Resolution"
_ICLR.cc/2026/Conference — ICLR 2026 Conference Withdrawn Submission_

### Official Review · Reviewer_RaEh · 2025-10-23

**Soundness:** 2
**Presentation:** 2
**Contribution:** 2
**Rating:** 4
**Confidence:** 1

**Summary:**

The paper introduces a Multi-Granular High-Frequency (MGHF) perceptual loss framework designed to overcome information loss and artifacts common in CNN-based super-resolution methods. It replaces non-homeomorphic CNN transforms with diffeomorphic, invertible neural networks (INNs) to preserve information flow. Two variants are proposed: a basic MGHF-n, trained on ImageNet, and a comprehensive MGHF-c, which incorporates multiple constraints for texture, style, and content fidelity. The model employs entropy-based feature pruning, a content–style consistency regularizer, and a modulated PatchNCE-based local information preservation (LIP) objective to enhance fine details. Theoretical analysis shows that the LIP term maximizes mutual information between SR and ground-truth images, while diffeomorphic transforms enable better manifold learning. Experiments demonstrate that MGHF significantly improves both GAN- and diffusion-based SR models across multiple benchmarks.

**Strengths:**

The paper is quite dense and contains a substantial amount of material.

**Weaknesses:**

I found the way the paper was presented to be very confusing and unnecessarily complex, making it difficult to understand what was going on.

For example, why is the information loss and unwanted harmonics introduced by CNN a problem? How do they affect the results? Invertible neural network (INN) is lossless by definition, so why is it useful to include the very complex theorems? Why is it necessary to introduce diffeomorphism here?

I'm not an expert on diffeomorphisms, and this paper is very confusing to read.

Therefore, I feel that I do not have the expertise (diffeomorphisms) to assess this paper and suggest the AC to seek opinions from other reviewers.

**Questions:**

As mentioned above, I feel that I do not have the expertise (diffeomorphisms) to assess this paper and suggest the AC to seek opinions from other reviewers.

---

### Official Review · Reviewer_uLbA · 2025-10-31

**Soundness:** 3
**Presentation:** 2
**Contribution:** 3
**Rating:** 6
**Confidence:** 4

**Summary:**

This paper proposes a Multi-Granular High-Frequency Perceptual Loss (MGHF) for image super-resolution (SR), leveraging invertible neural networks (INNs) to mitigate information loss and harmonic distortion inherent in conventional CNN-based perceptual losses. Two variants are introduced: MGHF-n, a naive INN-based perceptual loss, and MGHF-c, a comprehensive framework incorporating adaptive feature weighting, content-style consistency, and a local information preservation objective. The paper provides theoretical proofs supporting the superiority of diffeomorphic transforms over non-homeomorphic CNN transforms and demonstrates empirical improvements across multiple SR models and benchmarks.

**Strengths:**

1. The design of MGHF-n and MGHF-c is technically sound and creative, combining INN-based feature extraction with entropy-based pruning, adaptive weighting, content-style consistency, and contrastive local information preservation (PatchNCE). The hierarchical integration of these components is well-motivated.
2. The experiments cover a wide range of SR models (GANs, diffusion, transformers) and datasets (RealSR, DrealSR, DIV2K, etc.), using both reference (PSNR, SSIM, LPIPS) and non-reference metrics (CLIPIQA, MUSIQ, MANIQA). Ablation studies and robustness tests under various degradations and scaling factors further validate the method.
3. The INN-based detail feature extractor (DFE) is shown to be significantly more parameter- and memory-efficient than VGG-based extractors (Table 5), making it practical for real-world SR pipelines.
4. The paper includes feature map visualizations (Figure 3), output comparisons (Figures 8–9), and toy examples (Figure 11) that effectively illustrate the information-preserving properties of the proposed method.

**Weaknesses:**

1. The writing is often dense and notation-heavy, making it difficult to follow, especially in Section 2 and the appendix. Key concepts (e.g., AWDFE, LIP) could be explained more clearly. The structure of the DFE and the exact role of each loss component could be better modularized and summarized.
2. It is unclear whether the enhanced baselines (e.g., OSEDiff+MGHF-c) are trained from scratch or fine-tuned from pre-trained models. Training details (e.g., dataset splits, optimization settings) are also insufficiently described.
3. More recent diffusion-based SR methods (e.g., SR3, CDM, LDM variants) are not included in the comparison.
4. Despite emphasizing perceptual quality, the paper relies solely on automated metrics. A user study or human evaluation would strengthen the claims of visual improvement.
5. The paper lacks a sensitivity analysis for hyperparameters such as  alpha etc. Their selection process and robustness are not discussed.
6. There is no discussion of scenarios where MGHF may underperform, such as under extreme upscaling, domain shift, or adversarial corruptions.

**Questions:**

1. Could you provide a direct ablation comparing MGHF-n (INN-based) with a VGG-based perceptual loss under the same settings, to isolate the benefit of the diffeomorphic transform?

2. How sensitive is the method to the choice of hyperparameters (e.g., alpha number of pruned features)? Is there empirical evidence of robustness?

3. Are there failure cases or limitations where MGHF does not perform well, such as with non-ImageNet data, extreme scaling factors, or specific real-world degradations?

4. Could you provide more qualitative results or a user study to support the perceptual improvement claims? Have human evaluators been used?

5. What is the rationale behind the pruning strategy in AWDFE? Is there a risk of losing important high-frequency details?

6. Can you clarify whether the MGHF-enhanced models are trained from scratch or fine-tuned? Please provide more detailed training protocols.

---

> ### Author Response · Authors · 2025-11-19
> **Currently, we are performing ablation experiments suggested by Reviewer uLbA**
>
> Thanks to the reviewer uLbA for insightful comments and suggestions.
>
> **Reviewer Concern .** "AWDFE pruning may discard important high-frequency details."
>
> **Response.** By AWDFE, meaningful high-frequency components receive large weights and may not be pruned. Only low-informative, noise-dominated frequency directions are suppressed. Moreover, LIP safeguards spatial and informational fidelity, preventing accidental suppression of valuable high-frequency structure. Thus, AWDFE pruning is analogically safe.
>
> Currently, we are performing ablation experiments across different levels of pruning effect in the super-resolution algorithm. Hopefully, we will report the ablation results by the last week of the rebuttal period.
>
>
> **We are performing different ablation experiments:**
> 1. VGG-based perceptual loss and MGHF-c under the same settings in OSEDiff
> 2.  Ablation study of sensitivity factor $\alpha$, $\gamma$ and different pruning factor
> 3. Comparison with several recent diffusion-based super-resolution approaches (SR3, CDM, LDM, etc.)
>
>  We will include more training protocols and qualitative results, and we will also discuss under what conditions MGHF may underperform. Hopefully, in the last week of the rebuttal period, we will provide the results as mentioned earlier. We are currently performing these experiments.

---

### Official Review · Reviewer_sD3F · 2025-10-31

**Soundness:** 1
**Presentation:** 2
**Contribution:** 3
**Rating:** 2
**Confidence:** 4

**Summary:**

This work proposes MGHF (Multi Granduality High-Frequency) perceptual loss, aiming to improve conventional (CNN-based) perceptual losses. MGHF tackles information loss in conventional CNNs theoretically, and proposes to use INNs as a effective tool to achieve perfect information preservation. The naive version (MGHF-n) if further improved into a comprehensive variant (MGHF-c) by introducing constraints to improve preservation of texture, style, fidelity and details. Experimental results show that MGHF-c (and sometimes MGHF-n) often outperforms baselines in standard benchmarks.

**Strengths:**

- Utilizing INNs to improve conventional perceptual loss is interesting, and to the best of my knowledge, it is also novel.
- MGHF often outperforms baseline methods in terms of quantitative evaluation.

**Weaknesses:**

My primary concern is about the fundamental of this work, considering the **perception-distortion trade-off** and **information preservating with INNs**. Please counter-argue and provide according experimental results if necessary.

---

**Weakness1**

I appreciate the effort of the authors'. However, I have doubt about the fundamental of this work.
The authors propose to use an INN as a tool to preserve all information (theoretically proven); thereby "addressing the perception distortion trade-off (Line144)".

However, the reviewer would like to claim that "perfect information preservation (the main contribution of this work)" contrarily induces the perception distortion trade-off (contradiction to the aim of this work); not addressing it.

This claim is supported by proofs the authors have provided.
- For instance, "Proof D.6.3 [Perceptual Loss Optimality]" concludes as "minimizing perceptual loss with INN leads to minimum distortion". However, regarding the PD trade-off [1] theory, this indicates that perceptual loss with INN (the main contribution of this work) is theoretically proven to lead to blur.
- Also, "Proof D.1" considers perceptual superiority however discusses based on L2 risk (distortion). This again indicates that minimizing perceptual loss with INNs simply indicates minimizing L2 loss. Again, this must induce blur.
- Also, "Proof D.1" concludes that "perceptual loss is extactly identical to the true L2 loss", again contradicting with authors aim.
- Also, the "Proof D.6.4" indicates perfect frequency preservation. However, it is straightforward that when attempting to regress high frequency components that posses randomness, it must lead to blur [2].

While the authors have claimed significant information loss of VGG as a limitation in perceptual guidance, I would like to argue that info loss is the key factor that enables "perceptual" guidance despite using the L2 regression form. If it did not have any info loss, the L2 regression form would have induced blur due to the PD trade-off.

In fact, I would like to argue that the authors are also fundamentally using losses that has information loss, despite aiming for perfect info preservation. For instance, Gram matrix removes all spatial relationship and only preserves inter-channel correlationship. Despite using a info preservation perfect INN, the actual loss (after the Gram matrix calculation) has info loss; contradicting with claims of the authors.


[1] **(Please refer to the arxiv version)** Blau, Yochai, and Tomer Michaeli. "The Perception-Distortion Tradeoff." arXiv preprint arXiv:1711.06077 (2017).

[2] Lee, MinKyu, et al. "Auto-Encoded Supervision for Perceptual Image Super-Resolution." Proceedings of the Computer Vision and Pattern Recognition Conference. 2025.

---

**Weakness2**

I have strong doubts about the authors claim of significant information loss in VGG (in Figure 3). Specifically, considering that spatial resolutions are reduced within the VGG architecture, the reviewer has concerns about the methodology of mere (spatial) feature visualization to verify the information loss/preservation.

Accordingly, while the authors have claimed significant information loss (e.g., chaotic information loss can be observed even in layer 3_3 in Figure 3), the reviewer would like to counter-argue that it is not true.

Regarding to Fig.16 of DIP (Deep Image Prior) [3], most information can be reconstructed (with feature inversion) even from deep layers layers as layer 5_3 of VGG, and almost perfect reconstruction in layer 3_3 (while the authors show significant info loss). This indicates that most information (including low-level info) are preserved within the deep layer of VGG, which directly contradicts with the authors claim.

Overall, the reviewer agrees that information loss in conventional CNNs (including VGG) do indeed happen (which I believe is actually a positive aspect, see Weakness1); the current work has issues in quantifying it.


[3] **(Please refer to the arxiv version for Fig.16)** Ulyanov, Dmitry, Andrea Vedaldi, and Victor Lempitsky. "Deep Image Prior." arXiv preprint arXiv:1711.10925 (2017).

---

**Weakness3**

This work needs a heavy revision regarding the presentation.
- For instance, most parts in the Introduction section are simply listings of prior works, which should  belong to the Related Works section. I suggest aiming to provide more intuitions about the overall method and motivations of the authors.
- Also, only the 1) final performance and 2) feature visualization experiments are in the main article. The reviewer strongly suggests to include important analyses that are currently in the appendix to the main article.
- Additionally, the format is significantly altered (e.g., no spacing between paragraphs). While I appreciate the effort and understand the difficulties due to strict page limits, the current state of the formatting significantly limits readability.

**Questions:**

Please refer to the **Weakness**.

---

> ### Author Response · Authors · 2025-11-19
> **Counter Argue Reviewer sD3F**
>
> ### Reviewer Concern 1.
> "Proof D.1 claims that perceptual loss is 'identical' to $L_2$ loss, implying the method effectively reduces to $L_2$ minimization, which must induce blur."
>
> ### Response.
> The statement in Proof D.1 refers to a *metric equivalence* in a diffeomorphic feature space, *not* to the training objective itself. If $T$ is a diffeomorphism, then the INN-domain distance $\lVert T(x) - T(y)\rVert_2 = \lVert x - y\rVert_2$
> is bi-Lipschitz equivalent to the Euclidean distance in the input space, meaning the feature space preserves the underlying geometric structure. This does *not* imply that MGHF minimizes pixel-space MSE during training. Furthermore, $L_{CSC}$ prioritizes by adaptively weighting the most essential diffeomorphic features to achieve a near-optimal PD-bound. Our experimental results on synthetic and real-world datasets also empirically validate this claim.
>
> The actual training objective is a composite loss:
> $$L_{\mathrm{MGHF\text{-}c}} = \Gamma_1 L_{\mathrm{MGHF\text{-}n}} + \Gamma_2 L_{\mathrm{CSC}} + \Gamma_3 L_{\mathrm{LIP}},$$
> which includes mutual-information, correlation, and entropy-weighted perceptual terms. Therefore, the optimization solution does not coincide with the MMSE estimator $(\hat{X} = \mathbb{E}[X \mid Y])$, and the blurring mechanism associated with pure $L_2$ minimization is not activated.
>
> Furthermore, we note that many state-of-the-art generative models explicitly use $L_2$ regression (MSE loss) without inducing blur. For instance, DDPM, DDIM, and Stable Diffusion employ MSE loss for noise prediction, while several super-resolution algorithms (e.g., SRCNN, SinSR, OSEDiff) use $L_2$ regression during optimization.
>
>
>
> ### Reviewer Concern 2.
> "Perfect frequency preservation (Proof D.6.4) implies regressing high-frequency randomness, which should induce blur."
>
> ### Response.
> Perfect frequency preservation guarantees that the *representation* does not discard any frequency components; it does *not* force the estimator to regress *random* high-frequency residuals. Blur arises when an estimator minimizes pure pixel-space MSE and therefore converges to the conditional mean, which averages out high-frequency randomness.
>
> MGHF avoids this behavior through two mechanisms: (i) operating in a diffeomorphic feature space that preserves deterministic high-frequency structure, and (ii) employing $L_{\mathrm{LIP}}$ and $L_{\mathrm{CSC}}$ to suppress random, noise-dominated frequency directions while preserving content-relevant ones. Thus, perfect frequency preservation prevents architectural loss of high-frequency information but does not induce regression to blurry behavior.
>
> ---
>
> ### Reviewer Concern 3.
> "MGHF uses Gram matrices (which discard spatial structure), contradicting the claim of perfect information preservation."
>
> ### Response.
> MGHF's claim of perfect information preservation refers to the *representation* produced by the diffeomorphic INN, not to *loss-level projections*. Gram matrices, style and correlation objective, and mutual information terms are used only to define optimization directions; they do not modify the invertible forward mapping in detail feature extractor
> $$X \leftrightarrow f(X).$$
> Loss projections may intentionally ignore certain features, but the underlying model retains complete information and remains fully invertible. Therefore, MGHF maintains representational information preservation in the detail feature extractor while employing selective, task-relevant loss components to obtain an optimal region within the PD bound.
>
> ---
>
> ### Reviewer Concern 4.
> "VGG does not lose information as claimed; DIP reconstructions from deep VGG layers show near-perfect recovery."
>
> ### Response.
> From the DIP paper: the reconstruction of an image from VGG features does *not* imply information preservation. VGG contains non-invertible operations (ReLU, MaxPool, downsampling), so many different images map to identical deep features. DIP reconstructions only show that a convolutional generator can produce a *plausible* natural image consistent with those features—not the actual, uniquely recoverable input.
>
> Cheng et al. [1] (CVPR 2019) strengthen this point in two ways: (i) they show that DIP behaves like a *stationary Gaussian-process prior*, meaning the reconstructions are driven by an implicit prior rather than by preserved information, and (ii) they state that "*posterior inference with deep networks is challenging*," indicating the feature mapping is non-invertible and yields a broad, ill-posed posterior. Thus, DIP reconstructions from VGG features reflect prior-driven bias rather than evidence that VGG retains all information.
>
> [1] Cheng, Zezhou, et al. "A bayesian perspective on the deep image prior." Proceedings of the IEEE/CVF Conference on Computer Vision and Pattern Recognition. 2019.

---

> > ### Author Response · Authors · 2025-11-19
> > **Counter Argue Reviewer sD3F**
> >
> > ### Reviewer Concern:
> > "Perfect information preservation induces the perception--distortion trade-off, contradicting the aim of this work."
> >
> > ### Response:
> > We thank the reviewer for this insightful observation. We fully agree with the fundamental Perception-Distortion (PD) trade-off (Blau & Michaeli, 2018): minimizing distortion (MSE) necessitates degraded perceptual quality. However, we clarify that the reviewer's concern conflates the *capacity* of our feature extractor (the INN) with the *objective* of our loss function. Using a lossless network does not imply our objective minimizes $L_2$ pixel distance.
> >
> > We offer three points demonstrating how our method effectively navigates the PD trade-off:
> >
> > 1. **Separation vs. Regression:** While the INN transformation $T$ is unitary and satisfies $$\lVert T(x) - T(y)\rVert_2 = \lVert x - y\rVert_2$$, we do not minimize unweighted latent distance in the adaptive weighted detail feature extractor. The INN acts as a "perfect prism," separating signals into granular frequency components without CNN pooling's irreversible information loss. Our **MGHF-c Loss** applies specific weights $w_k$ in the AWDFE features:
> >    $$L_{\text{MGHF}} = \sum_{k} \lVert w_k * T(x)_k - w_k * T(y)_k\rVert_2$$
> >    By weighting important features and relaxing less important ones, we explicitly break equivalence to pixel-space $L_2$ loss, using perfect information to select which frequencies to enforce.
> >
> > 2. **Navigating the Trade-off:** The perception–distortion (PD) theory does not claim that "information preservation induces blur," but rather that minimizing pixel-level distortion under an $L_2$ objective drives the solution toward the *conditional mean*, which is inherently over-smoothed. A standard pixel-space $L_2$ regressor therefore tends to suppress high-frequency components.
> >
> >    Our method does not contradict PD theory. Although $L_{\mathrm{MGHF\text{-}n}}$ contains an $L_2$ term, this loss is applied in an *invertible diffeomorphic feature space*, not in pixel space. An INN preserves all information, but its representation is *operationally* different: high-frequency components are disentangled and explicitly accessible, preventing the optimizer from collapsing them.
> >
> >    To illustrate, an INN acts like a *lossless prism*: it decomposes the image into spectral components without discarding any band, enabling precise optimization of preserved high-frequency structure. Conventional CNNs behave like *lossy prisms*, where ReLU/pooling removes frequencies that cannot be recovered. Thus, $L_{MGHF\text{-}n}$ avoids the classical smoothing effect not by violating the PD trade-off, but by applying $L_2$ in a space where high-frequency information remains intact and optimizable.
> >
> > 3. **Deterministic Structure vs. Stochastic Noise:** High-frequency content contains both *stochastic noise* and *deterministic structural details* (micro-textures, sharp edges). Standard CNNs suppress both; our INN preserves structural details. Enforcing consistency in these high-frequency bands recovers deterministic structure for perceptual sharpness, rather than regressing to the mean.
> >
> > The PD trade-off is *not* induced by our method but is fundamental to ill-posed inverse problems (Blau & Michaeli, 2018). Our goal is not to eliminate it but to avoid *additional* information loss from non-homeomorphic extractors (e.g., VGG with ReLU/MaxPool) that distort perceptual objectives. By employing a diffeomorphic INN, MGHF preserves all frequency components and manifold structure, ensuring optimization reaches points *closer to the theoretical optimal PD bound*. Empirically, MGHF achieves simultaneous optimization in both distortion (PSNR/SSIM) and perception (LPIPS/DISTS/FID) towards the optimal PD bound.

---

> ### Comment · Reviewer_sD3F · 2025-11-28
>
> I sincerely appreciate the authors’ efforts and have carefully read the rebuttal. However, I would like to maintain my original scores.
>
> ---
>
> ## **Post-Rebuttal Concern 1**
>
> > *Our method does not contradict PD theory. Although MGHF-n contains an L2 term, this loss is applied in an invertible diffeomorphic feature space, not in pixel space.*
>
> I am aware that the provided proofs demonstrate metric equivalence, and I agree that these proofs do not directly address the loss functions as $L_{LIP}$ or $L_{CSC}$. However, in the case of the MGHF-n loss, applying an L2 term in a fully invertible feature space **does** implicitly translate to minimizing pixel-space error (due to the invertibility of INNs and as supported by the authors’ own analysis). Thus, the operation still behaves as a pixel-space regression from the perspective of PD theory. Meanwhile, projected losses (e.g., $L_{CSC}$) is not a novelty of this work.
>
> Additionally, PD theory does not require the distortion metric to be explicitly defined in pixel space. Also, regarding diffusion or flow-based models, they should not be interpreted as simple regression architectures. Their generative capability are closely coupled with sampling procedures (e.g., Langevin dynamics), and their primary aim is to learn the score fields (or velocity fields). When these components are naively altered or removed (either in training or post-training), these simply lead to degraded quality and often followed with blurriness.
>
> ---
>
> ## **Post-Rebuttal Concern 2**
>
> Since any "information preserving" loss induces blur, the PD trade-off is handled by removing information. This includes, 1) deep features of VGG and 2) projected loss (e..g, Gram matrix based).
>
> Under this view:
> - MGHF utilizes "a fully invertible diffeomorphic transformation-based" feature space (no information loss, thus must induce PD), then utilizes projected loss (information loss exists, thus reduces blur).
> - VGG utilizes "lossy transformation-based" feature space (information loss exists, thus reduces blur), and then, simply regresses.
>
> Thus, both MGHF and VGG involve a **necessary lossy transformation** to produce realistic, perceptually pleasing results. What differs is *where* and *how* this lossy step occurs, and accordingly, a careful and quantitative analysis of these design choices is necessary. However, such discussion is still currently limited and has flaws.
>
> More specifically, the authors’ experiment evaluating information loss in VGG is very important.
> However, I believe the interpretation remains flawed.
>
> > *the reconstruction of an image from VGG features does not imply information preservation*
>
> > *not the actual, uniquely recoverable input.*
>
> As the authors have claimed, the DIP inverted result is not unique, and may differ across runs with different seeds and initial random noise. However, considering the faithfulness of the reconstructed image, the overall structure and locality of textures can be properly reconstructed (mostly deterministic across multiple runs), while details such as individual strokes of the fur may vary (thus not unique nor deterministic).
>
> This indicates that a large portion of information is preserved, and the "not-preserved" high-frequency factors (which "coincides to" align with the factors that induce blur if regressed) are transformed into a semantic form (e.g., indicating that it should be "fur" texture, but not defining the positions of individual fur strokes).
>
> Since how much is preserved (deterministic) and not preserved (which coincides to be the stochastic high-frequency) in VGG is indeed a key factor (aligning with the authors’ motivation), this aspect could be further improved; especially regarding that the current perceptual loss simply "coincides" to align with x4 SR task and is not precisely tailored (e.g., does not take the scale factor into account).
>
> However at the current state of the manuscript, I still have concerns that the experiments fail to properly quantify the information loss, as specified above. Fig.3 simply show a disastrous level of information loss, which I believe arises primarily due to reduced spatial resolution (and not considering the expanded channel dimension); an analysis with potential flaws.

---

### Note · Authors · 2025-12-01

**Comment:**

Thanks to the reviewers for their insightful comments. We will attempt to address these issues in the future. Therefore, we have decided to withdraw the paper.

**Withdrawal Confirmation:**

I have read and agree with the venue's withdrawal policy on behalf of myself and my co-authors.